



# Influence of the North Atlantic Oscillation on annual spatio-temporal lightning clusters in western and central Europe

Markus Augenstein[1], Susanna Mohr[1,2], and Michael Kunz[1,2]

[1]Institute of Meteorology and Climate Research Troposphere Research (IMKTRO), Karlsruhe Institute of Technology (KIT), Karlsruhe, Germany
[2]Center for Disaster Management and Risk Reduction Technology (CEDIM), Karlsruhe Institute of Technology (KIT), Karlsruhe, Germany

**Correspondence:** Markus Augenstein (markus.augenstein@kit.edu)

**Abstract.** Based on lightning measurements in western and central Europe from 2001 to 2021 (May–August), a grid-based climatology and trend analysis of thunderstorm activity has been developed. The results indicate a significant decrease in thunderstorm activity in many regions. Extending the analysis beyond a purely grid-based approach, areas with spatio-temporal intense lightning (convective clustered events, CCEs) were identified in a second step by applying a clustering algorithm
(Spatio-Temporal Density-Based Spatial Clustering of Applications with Noise, ST-DBSCAN). For this purpose, a methodology is presented which seeks out to determine an appropriate density definition, as required by ST-DBSCAN.

An analysis of the characteristics of the CCEs indicates a slight increase of smaller, more separated clusters, while larger clusters occur less frequently over time. This suggests a shift in the mesoscale organization of convective systems. Furthermore, a correlation between the North Atlantic Oscillation (NAO) and thunderstorm frequency has been identified. Notably, there
was a pronounced reduction of thunderstorm activity, as well as an increased number of separated convective systems during negative NAO phases in France. This, in conjunction with a documented accumulation of years with predominantly negative NAO values between 2011 and 2020, is likely a contributing factor to the aforementioned negative trends.

## 1   Introduction

Thunderstorms and associated severe weather phenomena, such as heavy rain, hail, and convective wind gusts, frequently cause
harm to people and considerable damage to property as well as ecosystems across large parts of the world, including Europe (e. g., Holle, 2008; Ritenour et al., 2008; Kron et al., 2019; Púčik et al., 2019; Moris et al., 2020; Wilhelm et al., 2021). In the year 2023, for example, 59 % of all insured losses worldwide were attributed to severe convective storms (SCSs) (Swiss Re, 2024).

Given the increasing proportion of damage caused by SCSs, long-term changes in the frequency and intensity of these
events are of high relevance to various stakeholders such as the insurance industry or agriculture, as well as to society at large. Hoeppe (2016), for example, showed an increase in the annual number of insured losses from SCSs in Europe from 1980 to 2014. However, this trend is not solely attributable to meteorological trends; changes in vulnerability and assets also exert an important influence (Pielke, 2021).



Trend analyses of SCSs for the past are challenging mainly because of the transient and small-scale nature of thunderstorms. Furthermore, there is a lack of comprehensive and homogeneous direct observations of local thunderstorms and related phenomena over a sufficiently long-term period. As discussed in the latest Intergovernmental Panel on Climate Change Assessment Report (Seneviratne et al., 2023), findings of shifts in the frequency and/or intensity of such systems in the context of climate change are currently still surrounded by a high degree of uncertainty.

Several approaches or rather databases have been used in practice to estimate the thunderstorm frequency and activity over past decades: direct observations using (i) lightning data of different detection networks (Arrival Time Difference Network, ATDnet, Gaffard et al., 2008; EUropean Cooperation for LIghtning Detection, EUCLID, Drüe et al., 2007; Schulz et al., 2016; Poelman et al., 2016; lightning detection network, LINET, Betz et al., 2009), (ii) satellite data of overshooting tops (e. g., Christian and Latham, 1998; Cecil et al., 2014; Punge et al., 2017; Kaplan and Lau, 2021) or microwave radiometer data (e. g., Spencer et al., 1983; Laviola et al., 2020), or (iii) radar reflectivity (e. g., Nisi et al., 2016; Fluck et al., 2021). In addition, indirect measurements such as (iv) data from radiosoundings or reanalysis were used to identify convection-favoring atmospheric conditions based on parameters such as the convective available potential energy (CAPE), humidity in the lower troposphere, vertical wind shear, or combinations thereof (e. g., van Delden, 2001; Brooks et al., 2003; Kunz, 2007; Mohr et al., 2015; Rädler et al., 2018). Furthermore, convection-permitting regional climate models driven by reanalysis data could be used to analyze regional convection occurrence (e. g., Lucas-Picher et al., 2013; von Storch et al., 2017). However, thunderstorm simulations, even performed with convection-resolving numerical weather prediction or regional climate models, are accompanied by a high degree of uncertainty, mainly due to the complex interactions between dynamic processes and related thermodynamic effects. While direct observations (i-iii) are only available for a limited period of time, which in most cases doesn't allow for the derivation of reliable trends, indirect data are available for several decades, allowing for robust trend analysis.

Almost all studies based on indirect data have found an increase in thunderstorm potential over large parts of Europe, mostly due to an increase in lower tropospheric humidity in recent decades (e. g. Mohr and Kunz, 2013; Sanchez et al., 2017; Rädler et al., 2018; Ghasemifard et al., 2024). For example, Battaglioli et al. (2023), for example, found that there was a significant increase in (modeled) lightning and (large) hail in most parts of Europe since the 1950s, with the largest increase in hail occurring in northern Italy. The so-called ingredient-based forecasting, however, can only estimate the convective potential of the atmosphere and does not allow conclusions regarding actual SCSs. Groenemeijer et al. (2017), for example, found that in Europe SCSs only occur on 60 % to 80 % of all days with sufficient high CAPE and wind shear. Taszarek et al. (2019) compared different thunderstorm climatologies based on several data sources (lightning, reports, soundings, and reanalysis) for Europe and found noticeable differences between the climatologies depending on the data source. This illustrates the uncertainty scientific research is confronted with when pursuing to detect and analyse thunderstorm activity.

Furthermore, large-scale atmospheric processes and mechanisms exert a controlling element concerning thunderstorm activity, thereby affecting trends. Piper and Kunz (2017) and Piper et al. (2019), for example, found a relation between lightning activity and European teleconnections, such as the North Atlantic Oscillation (NAO), the East Atlantic and the Scandinavian patterns. They attributed these variations to the large-scale forcing and to anomalies in the sea-surface temperatures (SST)



leading to the development of thunderstorms. In addition, some studies suggest a relationship with specific weather patterns such as atmospheric blocking (e. g., Mohr et al., 2019, 2020; Ibebuchi, 2022).

Most studies that estimate the convective climate or assess trends in convective potential employ a grid-based assessment of SCSs, which does not capture the spatio-temporal behavior of thunderstorms across several grid points. Most thunderstorms, especially those with high damage potential, occur not only as isolated events, but also in the form of larger convective clusters such as mesoscale convective storms or complexes with a spatial extent of several hundred kilometers (Schumacher and Rasmussen, 2020). The characteristics of these larger events cannot be captured by the grid-based approach. For this purpose, clustering methods such as the Density-Based Spatial Clustering of Application with Noise (DBSCAN; Ester et al., 1996) and similar other methods have been successfully applied to the analysis of thunderstorms based on lightning strikes (e. g., Hutchins et al., 2014; Galanaki et al., 2018; Pérez-Invernón et al., 2021; Shi et al., 2022; Hayward et al., 2023). The method ST-DBSCAN (Spatio-Temporal Density-Based Spatial Clustering of Applications with Noise) extends the DBSCAN algorithm by a temporal dimension and generally identifies clusters in a point cloud based on the spatial-temporal density of the points themselves. This algorithm does not rely on man-made metrics (such as grids), but objectively identifies clusters of any shape. This is a major advantage over other clustering algorithms like, for example, k-Means, which need additional information such as the number of clusters to be detected (MacQueen et al., 1967).

So far, this approach has generally been used in connection with lightning strikes to distinguish between individual thunderstorms. For example, the ST-DBSCAN was applied to estimate the global electric circuit activity (Hutchins et al., 2014) or to determine thunderstorm tracks (Hayward et al., 2023). The application of the algorithm to lightning strikes allows for the identification of a spatio-temporal area of high convective activity. This in turn enables the derivation and more detailed analysis of characteristic properties such as size, duration, and shape of the convective clusters.

The main objectives of our study are: (i) to identify convectively active regions of coherent thunderstorms and to analyze their spatio-temporal behavior; (ii) to present a grid-based, high-resolution climatology and trend analysis of thunderstorm frequency based on direct lightning measurements for western and central Europe over a 21-year period; (iii) to investigate the temporal distribution of lightning clusters detected by ST-DBSCAN; and (iv) to establish possible connections of the convective clusters with teleconnection patterns (here the NAO) and their influence on the temporal variability of the clusters.

The paper is structured as follows: Section 2 gives an overview of the data sets and used methods. Section 3 demonstrates the application of the ST-DBSCAN to lightning data in order to identify CCEs and, additionally, explains the derivation of suitable density parameters for the ST-DBSCAN. Section 4 presents a grid-based climatology of thunderstorm days and lightning density, followed by trend analyses of both quantities with high resolution. In Section 5 the spatio-temporal characteristics of CCEs are examined. Finally, the relationship between convective clustered events and the NAO is discussed in Section 6. Section 7 discusses and summarizes the primary findings, which are then used to support the formulation of conclusions.





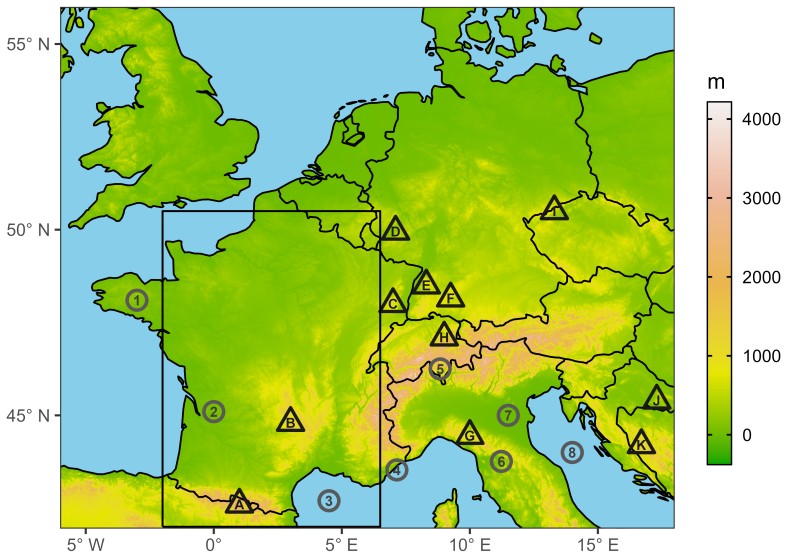

**Figure 1.** The topography of the investigation area is shown, as are the geographical regions (numbers in circles) and mountainous regions (capital latin letters in triangles) that are referenced in this study: 1 Brittany, 2 Gironde estuary, 3 Western Mediterranean Sea, 4 Nice, 5 Ticino, 6 Florence, 7 Po Valley, 8 Adriatic Sea; A Pyrenees, B Massif Central, C Vosges, D Eifel, E Black Forest, F Swabian Jura, G Appenines, H Säntis, I Ore Mountains, J Pozegan Mountains (Psunj), K Dinaric Alps. The black box indicates the sub-study area mentioned in Section 6.

## 2 Data and methods

The spatio-temporal variability of SCSs and its relation to large-scale patterns is investigated for the period between May to August (MJJA) from 2001 to 2021, as most thunderstorms in western and central Europe are observed during this season (Wapler, 2013; Anderson and Klugmann, 2014; Poelman et al., 2016; Piper and Kunz, 2017; Taszarek et al., 2019). The investigated area ranges from $6°$ W to $18°$ E and $42°$ N to $56°$ N, covering most parts of western and central Europe (Fig. 1).

### 2.1 Lightning data

Cloud-to-ground (CG) strokes from the ground-based low-frequency lightning detection system EUCLID (Drüe et al., 2007; Schulz et al., 2016; Poelman et al., 2016) were used in this study. Lightning strikes are formed by multiple lightning stokes (typically around 3; Rakov, 2016). As discussed in more detail in Schulz et al. (2016), several changes and upgrades in the sensor technology and the detection algorithm were made during the 21-year period under investigation. For example, the median location accuracy of CG strokes was improved from over $300\,\mathrm{m}$ to less than $90\,\mathrm{m}$ (Schulz et al., 2016). Increased precision regarding the location accuracy (in the order of magnitude of several hundred meters) has a negligible impact on the determination of so-called thunderstorm days (TDs; see Sect. 4.1) and on the composition of the convective clustered





events derived from applying ST-DBSCAN (see Sect. 3). For statistical consistency and to ensure that the data set is nearly homogeneous and robust over the whole study period, CG strokes with weak positive peak currents of less than $10\,\mathrm{kA}$ were

105 excluded from the data set based on the recommendation of the network operator in Germany (Thern, 2020). This was done because research has shown that lightning strokes with weak positive peak currents are more likely to be classified as cloud-to-cloud (CC) lightning (Cummins et al., 1998; Wacker and Orville, 1999a, b) and therefore, in the years after 2015, the classification algorithm used by EUCLID classifies weak positive flashes almost exclusively as CC strokes.

In total, 87 % of all CG strokes in the summer half-year (2001–2021) occur between May to August. April and September

were excluded from the analysis due to high annual seasonal variability and the fact that the ratio of thunderstorm activity over land to that over the Mediterranean decreases in September because of the relatively higher sea surface temperatures, which complicates geographical comparisons (cf. Piper and Kunz, 2017).

Using an objective method, Piper and Kunz (2017) determined a TD when at least five CG strokes were observed within an area of $10 \times 10\,\mathrm{km}^2$ on the same day. This definition was also applied in our study. A binary measure (TD yes/no) used for

climatological observations and trend analyses has the advantage that it is not dominated by single severe thunderstorms as it might happen when using lightning number or density directly. Because a single severe thunderstorm event can produce an enormous number of lightning flashes in a spatially and temporally limited area, they can distort statistical considerations by overweighing individual storms.

## 2.2 North Atlantic Oscillation

Low-frequency modes of climate variability of the Northern Hemisphere are described by so-called teleconnection patterns. Natural climate variability is reflected to a large part by variations in teleconnection patterns. The leading mode on the Northern Hemisphere, the NAO, is particularly relevant for Europe. To investigate the relationship between the NAO and the frequency and type of thunderstorms, monthly values of the NAO index were considered to identify years (here MJJA) with unusual accumulations of negative or positive NAO values ($Y_{\mathrm{NAO}-}$ or $Y_{\mathrm{NAO}+}$, see Sect. 6 for an exact definition). These data are made

available by the US National Oceanic and Atmospheric Administration since 1950. The index is derived via a rotated S-mode principal component analysis (Richman, 1986) applied to monthly mean standardized $500\,\mathrm{hPa}$ geopotential height anomalies (Barnston and Livezey, 1987) which are provided by the National Centers for Environmental Prediction/National Center for Atmospheric Research Reanalysis 1 (NCEP-NCAR1; Kalnay et al., 1996).

## 2.3 Trend analysis

Non-parametric linear trend analysis was used to investigate systematic changes in the time series $y_t$ with the time steps $t$ (here the years) and $y$ the annual number of TDs or CG strokes. Non-parametric methods are generally less powerful than parametric trend analysis (such as, for example, the minimization of the sum of squared errors, Abdi, 2007), but are less affected by outliers (Lanzante, 1996). To calculate the slope of the time series $\hat{\beta}$, the repeated median estimator (RM; Siegel, 1982), an





extension of the Theil-Sen estimator (TS; Theil, 1950; Sen, 1968), was used:

$$\hat{\beta}_{\mathrm{RM}}(t,y) = \mathrm{med}_i \, \mathrm{med}_{j \neq i} \frac{y_j - y_i}{t_j - t_i} \, . \tag{1}$$

As outlined in Eq. 1, the median slope is calculated for each time step, and subsequently, the median of all the median slopes of all time steps is determined for the final estimator, denoted as $\hat{\beta}_{\mathrm{RM}}(t,y)$. The RM exhibits a superior breakdown value in comparison to the TS, thereby providing a more robust estimator of $\hat{\beta}$.

The significance of the trend in the time series $y_t$ is calculated using the non-parametric robust Mann-Kendall test (Mann, 1945; Kendall, 1975). However, the Mann-Kendall test is susceptible to autocorrelation in time series, which can lead to an increased false rejection of the null hypothesis that no trend exists in the series. To address this limitation, Yue et al. (2002) therefore proposes a four-stage process to mitigate the potential issues associated with the Mann-Kendall test (trend-free pre-whitening, TFPW):

1. Estimation of $\hat{\beta}$ using the TS-method from the unprocessed original data.

2. Removal of the trend (detrend, detr) from the time series $y_t$:

$$y_{t, \, \mathrm{detr}} = y_t - \hat{\beta} t \, . \tag{2}$$

3. Removal of the autoregressive lag-1 process (correlation between values of the time series of one time step) from $y_t$, so called pre-whitening (PW):

$$y_{t, \, \mathrm{PW}} = y_t - \mathrm{AR}(1)(y_{t-1}) \, . \tag{3}$$

4. Addition of the trend to $y_{t, \, \mathrm{PW}, \, \mathrm{detr}}$:

$$y_{t, \, \mathrm{TFPW}} = y_{t, \, \mathrm{PW}, \, \mathrm{detr}} + \hat{\beta} t \, . \tag{4}$$

Finally, the TFPW time series $y_{t, \, \mathrm{TFPW}}$ is evaluated for statistical significance through the application of the Mann-Kendall test. While the TFPW process enhances the test power, this is accompanied by an increased likelihood of erroneously rejecting the null hypothesis (Yue et al., 2002).

## 2.4 ST-DBSCAN

One widely used density-based clustering algorithm of point data is the DBSCAN (Ester et al., 1996) or its extension in the temporal dimension, the ST-DBSCAN (Birant and Kut, 2007). This algorithm clusters points based on a predefined density criteria: A minimum number of points (here CG stokes, *minPts*) within a certain area ($\varepsilon_{space}$) and within a certain time ($\varepsilon_{time}$).

In mathematical terms, a point within a point cloud $P$ is classified as 'dense' if the quantity of points, denoted as $q$, within the so-called $\epsilon$-environment,

$$N_\epsilon(p) = \{q \in P \, | \, dist_{\mathrm{space}}(q,p) \leq \epsilon_{\mathrm{space}} \, \wedge \, dist_{\mathrm{time}}(q,p) \leq \epsilon_{\mathrm{time}}\} \, , \tag{5}$$



exceeds *minPts*. In this study, the $dist_{space}(q,p)$-function is the Euclidean distance, although other distance metrics could be used. The $dist_{time}(q,p)$ function simply describes the temporal difference between two points. As an example, Fig. 2a shows the (two-dimensional, 2D) $\epsilon$-environment of each point illustrated as circles.

Based on the above described criteria, the algorithm categorizes the point data into three groups: core points, border points, and noise (Fig. 2a). Border points ensure that the core points meet the density criteria, but they do not meet it by themselves. The final clusters are thus constituted of core and border points. Once a density definition has been set, ST-DBSCAN identifies clusters of any arbitrary shape, as shown exemplary in Fig. 2b for lightning strikes on one day. Note the temporal differentiation of clusters in Italy (red and green).

The methodology employed to determine reasonable density parameters for the ST-DBSCAN algorithm in three-dimensional (3D) space (see Sect. 3 for development) is similar to that utilized to determine 2D density criteria. The heuristics used to determine suitable parameters for the 2D DBSCAN algorithm are ($k$NN distance approach; Ester et al., 1996):

1. The distance of each point to its $k$-th neighbor (nearest neighbor, $k$NN) is calculated.

2. The distances determined are sorted in ascending order.

3. On the basis of the graph of the function, a visually identifiable sudden increase in the $k$NN-distance (the so-called 'knee'), indicates the optimum value for $\epsilon_{space}$ (see Fig. 2a, bottom right).

In the case of the 3D algorithm ST-DBSCAN, there is no mathematically defined procedure, as the additional time dimension makes it impossible to calculate a single distance. Therefore, in context of lightning strokes, a variety of approaches dealing with this matter exist in the literature: For example, Hutchins et al. (2014) had the aim to identify single thunderstorms and

used *minPts* = 2, $\varepsilon_{time}$ = 18 min and $\varepsilon_{space}$ = 0.12° (approx. 13 km) as parameter values. Pérez-Invernón et al. (2021) on the other hand suggested a shorter time ($\varepsilon_{time}$ = 15 min) than Hutchins et al. (2014) in order to discriminate better between single and isolated thunderstorms in close succession in the same area. Both used *minPts* = 2, so even clusters with only two lightning strokes are considered. A different approach was taken by Shi et al. (2022), whose intention was to modify the parameters to achieve the greatest possible ratio of clustered to non-clustered flashes ($\varepsilon_{space}$ = 0.10°, approx=11 km; $\varepsilon_{time}$ = 15 min).

The brief discussion above illustrates that a precise definition of the parameters always depends on the research objective for which the method is ultimately used. In this regard a scientific approach is preferable to an arbitrary definition of the individual parameters. The methods mentioned so far refer to individual convective systems (including weak ones such as single cells). However, the purpose of this study is to identify contiguous areas of intense thunderstorm activity, which may also consist of several thunderstorm systems. Therefore, a novel heuristic was developed (see Sect. 3) to determine optimal parameter values

similar to the above mentioned 2D $k$NN distance approach for DBSCAN.

## 2.5 Bagplot

Bagplots are used to assess the distribution of the polygons of the convective clustered events in spatial and temporal dimensions (Rousseeuw et al., 1999). A bagplot is a bivariate generalization of a boxplot in two dimensions visualizing the location, spread,





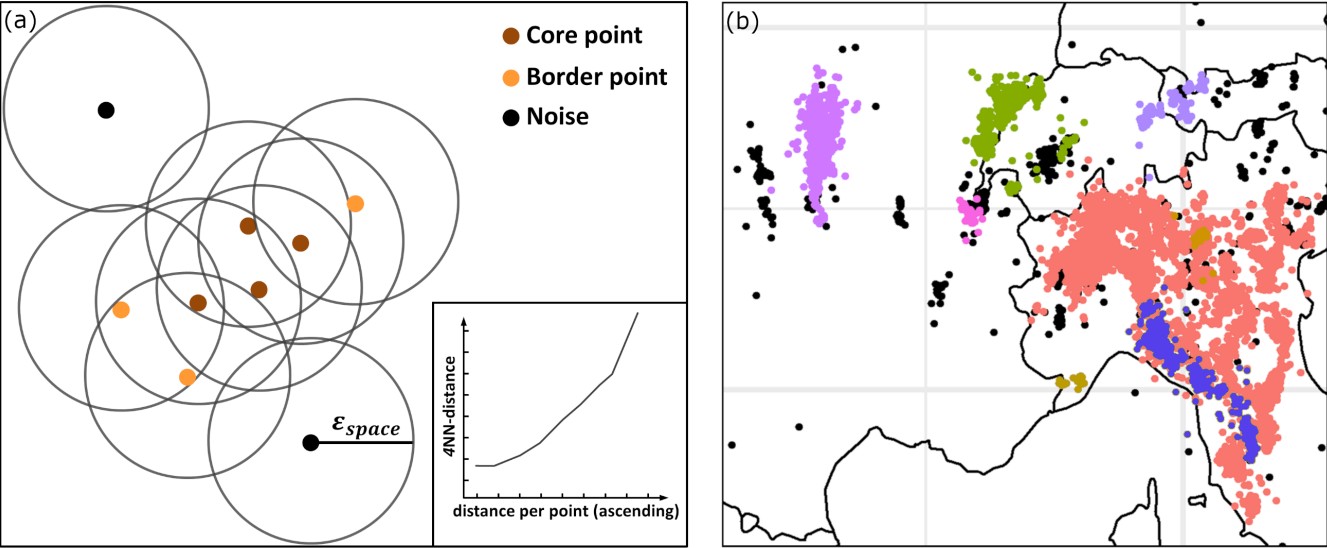

**Figure 2.** (a) Schematic illustration of how the DBSCAN algorithm distinguishes points from a point sample into core points (brown), border points (orange), and noise (black) based on predefined density criteria (here exemplarily $minPts = 4$; black circles indicate $\varepsilon_{\text{space}}$). The $k$NN-distance (here exemplarily $k = 4$, see definition in text) of each point sorted in ascending order is shown in the bottom right. (b) Example of clusters (different colors) and noise (black) derived with ST-DBSCAN from EUCLID lightning data on 9 May 2001 ($minPts = 40$, $\varepsilon_{\text{space}} = 50\,\text{km}^2$, $\varepsilon_{\text{time}} = 20\,\text{min}$). Note also the temporal differentiation of clusters in Italy (right, red and blue clusters). See Section 3 for the definition of the density parameters used in the study.

correlation, skewness, and tails of the data (Rousseeuw et al., 1999). The center is called the depth median and is equivalent

to the median in one dimension. A bag containing 50 % of the data points surrounding the depth median corresponds to the interquartile range. A second convex hull is constructed by inflating the outer bag by a selectable factor (default: 3). In the following, a factor of 6 is more practical as it sharpens the visual distinction between the inner and the outer bag (see Sect. 5.2). The convex hull enveloping the data points of the loop corresponds to the whiskers in a one-dimensional boxplot. Outliers are data points on the outside. Rousseeuw et al. (1999) provides a more detailed explanation, and Hyndman and Shang (2010)

describe the implementation in the programming language R.

## 2.6 Odds ratio

A statistical parameter to describe the degree of correlation between two binary variables is the odds ratio (OR). The odds are the probability that an event occurs ($P$) divided by the probability that the event will not occur ($1 - P$; Schuchard-Ficher et al., 2013). In this study, the OR quantifies, for example, the statistical relation between the binary variables TD and the years with

an unusual accumulation of months with negative NAO values ($Y_{\text{NAO}-}$). The OR is calculated as the ratio of the conditional probability (here the relative frequency) of the occurrence of an event (here TD) when another event (here $Y_{\text{NAO}-}$) is present



to that without the event (here all other years except of $Y_{\text{NAO-}}$):

$$OR = \frac{\frac{P(\text{TD}|Y_{\text{NAO-}})}{1-P(\text{TD}|Y_{\text{NAO-}})}}{\frac{P(\text{TD}|(1-Y_{\text{NAO-}}))}{1-P(\text{TD}|(1-Y_{\text{NAO-}}))}} = \frac{P(\text{TD}|Y_{\text{NAO-}})}{1-P(\text{TD}|Y_{\text{NAO-}})} \cdot \frac{1-P(\text{TD}|(1-Y_{\text{NAO-}}))}{P(\text{TD}|(1-Y_{\text{NAO-}}))} . \tag{6}$$

Thus, the OR is interpreted as follows:

OR>1: The odds of TDs in $Y_{\text{NAO-}}$ is larger and the probability of more TD during $Y_{\text{NAO-}}$ is higher.

OR<1: The odds of TDs in $Y_{\text{NAO-}}$ is smaller and the probability of more TD during $Y_{\text{NAO-}}$ is lower.

OR=1: There is no difference in the TD occurrence between $Y_{\text{NAO-}}$ and those without $1-Y_{\text{NAO-}}$.

To determine the significance of the OR values between the two variables TD and $Y_{\text{NAO-}}$, Fisher's exact test was used (Fisher, 1992). This test is particularly suitable when the sample sizes are small (Fisher, 1992), which is the case in regions where TDs 215    are climatologically rare events.

## 3    Identification of convective clustered events

In the context of this study, the ST-DBSCAN algorithm generates so-called convective clustered events (CCEs). These are defined as temporally and spatially contiguous areas of high convective activity and are computed with the ST-DBSCAN algorithm. The CCEs are further analyzed with respect to possible changes in their duration and spatial extent. This section 220    begins with an introduction to the heuristics used to determine appropriate density-defining parameter values.

### 3.1    Selecting density parameters for the ST-DBSCAN approach

To find appropriate values for the density-defining parameters *minPts*, $\epsilon_{\text{time}}$, and $\epsilon_{\text{space}}$, the 2D $k$NN distance approach (see also Sect. 2.4) is further developed for the 3D space. The basic 2D procedure is briefly explained here using a fictitious case as an example (Fig. 3): The parameter $\epsilon_{\text{space}}$, which takes into account the distance between two CG strokes, is increased successively 225    and for each $\epsilon_x$ (with different spatial distances $x$), Fig. 3 shows the percentage of the lightning stroke sample which enclose at least *minPts* (here 4). The figure also shows the percentage of points considered as dense at different distances. The resulting graph is similar to the inverse function of the $k$NN distance function (see Fig 2). However, the decisive advantage of this procedure in contrast to the classic $k$NN distance approach is that the temporal (third) dimension $\epsilon_{\text{time}}$ can also be successively extended.

In the following, we now transfer the 2D method to 3D. Assuming a fixed value for *minPts*, we vary $\epsilon_{\text{time}}$ and $\epsilon_{\text{space}}$ in small increments and examine the resulting percentage of CG strokes fulfilling the density criteria as described in Fig. 3. For this purpose, we take a random sample of our CGs. In the example, we choose a fixed *minPts* and for each CG stroke, we examine different combinations of parameters and check whether the density criteria is fulfilled (namely reach *minPts* in the chosen $\epsilon_{\text{space}}$ and $\epsilon_{\text{time}}$). As a result, Fig. 4 shows the (interpolated) percentage of clustered CG strokes of the CG stroke sample 235    from all years. The sample contains 750 CG strokes per year. Sensitivity studies for other sample sizes showed no significant changes to the results. As with the $k$NN distance approach, it should be noted that this procedure does not determine a fixed, unambiguous numerical value, but rather a range of values that proves to be optimal for separating noise and cluster points.





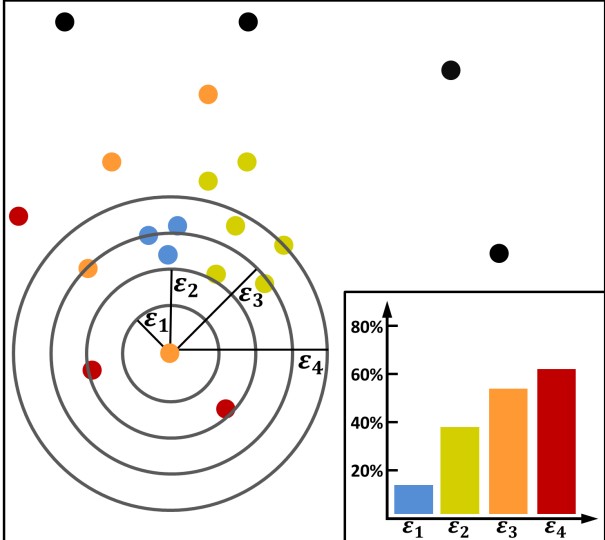

**Figure 3.** Exemplary sketch for determining 'optimal' $\epsilon_{space}$ values for the ST-DBSCAN algorithm using the 2D modified $k$NN distance approach (detailed definition in text). Point sample (here exemplarily 20 points) and different $\epsilon_1$ to $\epsilon_4$ around a starting point (black circles). Different colors indicate points categorized as dense with different density definitions: *minPts* = 4 and $\epsilon_x$ ($\epsilon_1$ red, $\epsilon_2$ orange, $\epsilon_3$ light green, $\epsilon_4$ light blue). Black points indicate noise (not matching the density criteria regardless of $\epsilon_x$). The bottom right corner shows the percentage of points that meet the density criteria as a function of $\epsilon_x$.

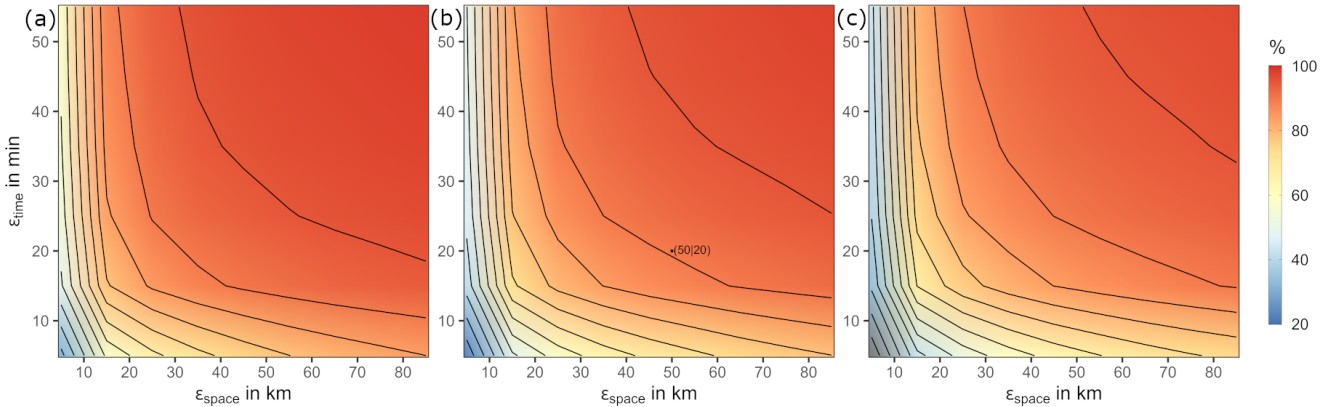

**Figure 4.** Similar to Fig. 3, but with the addition of a temporal dimension (bottom right): Shown is the (interpolated) percentage of a sample of CG strokes using the ST-DBSCAN (here 750 strokes, for details see text) as a function of the three spatio-temporal density definitions: different minimum number of points *minPts* (a) 30, (b) 40, and (c) 50 and within a certain area ($\varepsilon_{space}$) and within a certain time ($\varepsilon_{time}$) and to the color gradient associated contour lines (black lines). The black dot marks the final chosen values in the study, of the two $\epsilon$-environments $\epsilon_{space}$ (50 km) and $\epsilon_{time}$ (20 min) with *minPts* = 40.





Figure 4 shows the percentage of CG strokes classified as dense with different numbers of *minPts* (from left to right: 30, 40, 50) within the spatio-temporal $\epsilon$ environment. The figures have an additional (here colored) dimension compared to Fig. 3. The results prove to be robust with regard to variations in *minPts*. In general, all subfigures in Fig. 4 show a pronounced inverse 3D-'knee' of the distribution (indicated by the strong changes in the gradient over a relatively small area). This shifts with increasing size from *minPts* to larger spatial and temporal $\epsilon$ values. It can also be seen that for values of $\epsilon_{\text{space}} < 30\,\text{km}$ changes in $\epsilon_{\text{time}}$ have little influence on the distribution of the gradient. In the opposite case, i. e., small values of $\epsilon_{\text{time}} < 20\,\text{min}$, changes in $\epsilon_{\text{space}}$ have a stronger influence on the gradient. This fact can be observed independently of *minPts*. Using this method, it is now possible to specify a range of values $\mathbb{W}$ for the meaningful parameters $\epsilon_{\text{space}}$ and $\epsilon_{\text{time}}$: $\mathbb{W}_{\epsilon_{\text{space}}} = [20, 80] \wedge \mathbb{W}_{\epsilon_{\text{time}}} = [15, 25]$. There is only a slight dependence on *minPts*, meaning that $\mathbb{W}$ is relatively independent of the required minimum number of CG strokes.

The final density-defining parameters selected for further investigation in this study were determined based on a visual assessment of the clusters formed by different parameter combinations of this value range. As this study focuses on the identification of CCEs, where several thunderstorms are in close spatial and temporal relation to each other, we set the threshold to *minPts* = 40 (Fig. 4b). This value of *minPts* = 40 proved to be a good compromise between the identified area size and avoiding too many isolated thunderstorms. Fig. 4a and c show that even changes of *minPts* $\pm$ 10 have negligible effect on the results using *minPts* = 40.

As described by Ester et al. (1996), border points actually can be assigned to two clusters simultaneously. However, a border CG stroke is only assigned to one cluster by the algorithm. Therefore, in some rare cases, a certain cluster, whose boundary points have already been assigned to the neighboring cluster, can have fewer than 40 CG strokes. These clusters with less than *minPts* = 40 are removed from the data set.

## 3.2  Determination of CCEs by applying ST-DBSCAN to lightning data

The application of the ST-DBSCAN algorithm to the lightning data categorizes each individual CG stroke (as long as it is not defined as noise) into a CCE. The spatio-temporal localization of the CCEs is determined by a polygon, which is defined by CG strokes on the edge of the cluster. The so-called alpha shapes method (Edelsbrunner et al., 1983) is used to determine suitable polygon envelopes of the individual clusters. Polygons, whose internal angles are always smaller than $180\,°$, have no 'indentations' and are referred to as convex (otherwise as concave). The number of 'indentations' is determined by a parameter that defines the so-called concavity (see Fig.5a, b, c). The purpose in the following is to achieve a balance between the enveloped area that describes the convective area and the smallest possible number of 'indentations' when creating the polygons. In visual terms, the concavity parameter modulates the strength of a rubber band spanning the points (see different results for varying concavity values in Fig. 5a, b, c).

A suitable concavity parameter value optimizes the balance between spatial accuracy and reasonable enveloping. To determine an optimal threshold, the ratio of the respective enclosed area to an area resulting from a concavity parameter value of 1 was calculated. This optimization problem is highlighted in Fig. 5d, which shows the mean of the area ratio of all detected CCEs (black line) including the 75 % and 25 % percentiles as spread (gray area). The mean area ratio increases sharply with



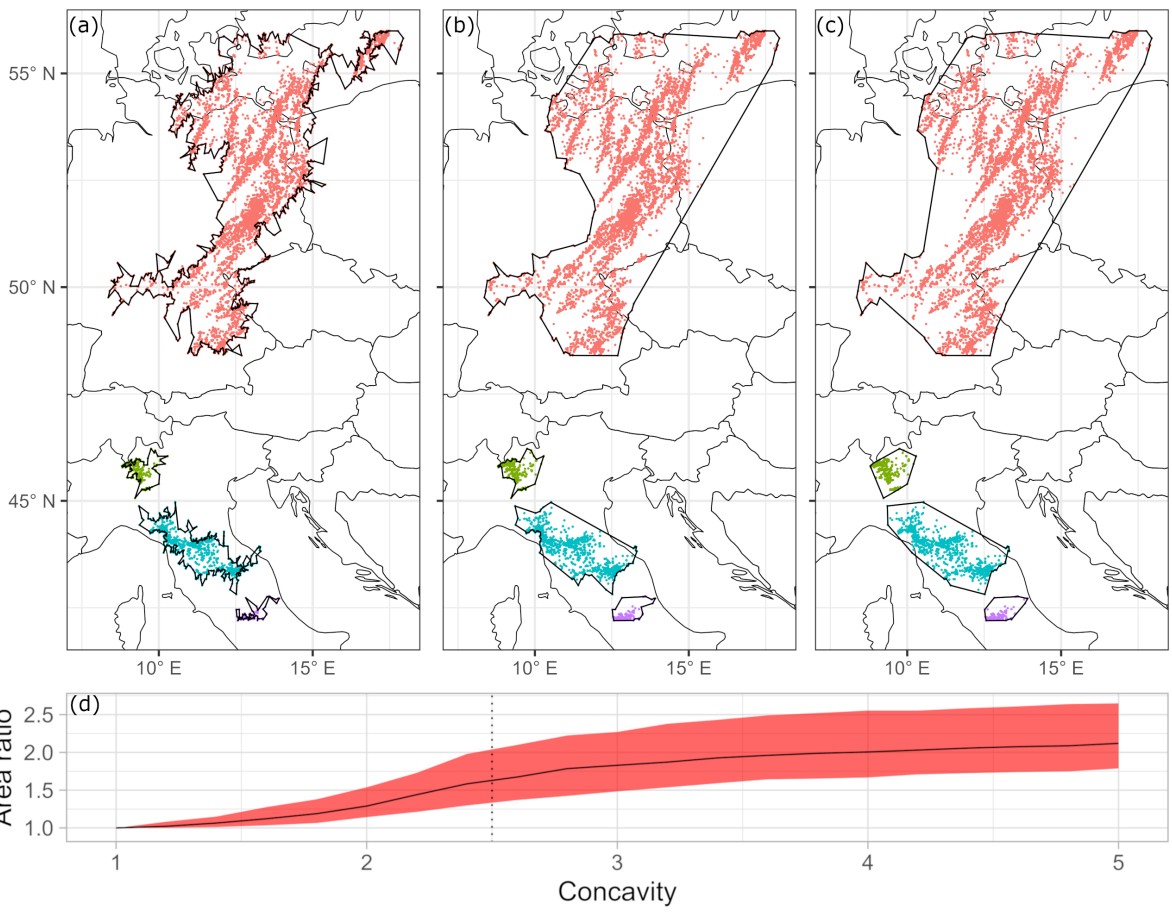

**Figure 5.** Different examples of the resulting area of a CCE using ST-DBSCAN depending on the choice of the concavity parameters that define the respective envelope of the CCE: (a) 1, (b) 2.5 (used in this study), and (c) 5. (d) Ratios of the area size (respective concavity parameter to concavity parameter = 1) of all CCEs (based on EUCLID, 2001–2021, MJJA) as a function of the concavity parameter (mean as black line, 25 % and 75 % percentiles in gray); the vertical line indicates the final value of 2.5 used in the study.

increasing concavity to a value of about 2.2. The 75 % percentile shows this strong increase until a value of about 2.4 is reached. For the 25 % percentile, this happens for slightly lower values around 2. After that, the slope decreases for the mean and both percentiles, showing that the increase in area decreases from there, even as the concavity parameter increases. As a suitable
value we choose 2.5 (vertical dotted line in Fig. 5d), just after the rates of change become negative (see Fig. 5b).



## 4 Climatology and trends in thunderstorm activity

To investigate spatial differences and trends of thunderstorms in western and central Europe with a high resolution, the mean annual number of TDs (definition see Sect. 2.1) and the mean annual lightning density (number of CG stokes in a $10 \times 10\,\mathrm{km}^2$ area) are considered.

### 4.1 Climatology

Both, TDs and the annual CG stroke density, show an increase from the coast to the inland on the continental scale (Fig. 6). This increase partly results from the climatological distribution of atmospheric stability (Mohr and Kunz, 2013). Mean annual numbers of TDs of 1 or less occur over large parts of Great Britain and Brittany in France with less than 20 mean CG strokes per year. More than 12 days per year with up to 500 CG strokes can be observed around the southern slopes of the Alps. By 285 contrast, the highest elevations of the Alps are an area of particularly low CG and TD numbers, which was already found in analyses of lightning (Manzato et al., 2022) and overshooting top data (Punge et al., 2017; Giordani et al., 2024). The grid point with the highest number of CG strokes in the study area (near the border to Austria; see Fig. 1 H) contains Mount Säntis in Switzerland. An exposed TV antenna on the mountaintop is the reason for the very high lightning frequency of over 500 CG strokes per year on average (Manoochehrnia et al., 2008).

Comparing the spatial distribution of TDs (Fig. 6a) with the CG stroke density (Fig. 6b), most of the hot spots (local and total) are identical since both quantities are highly correlated (Spearman rank correlation coefficient: r = 0.73, p < 0.05). The most prominent exception of this relationship is the northern part of the Adriatic Sea, where the number of TDs and CG stroke density diverge, resulting in a higher number of lightning strokes per TD. In this area, an unusually high amount of lightning strokes per single TD is found (not shown). A reason for this could be the more stationary and isolated thunderstorm systems 295 that preferably occur over the Adriatic Sea (Kotroni and Lagouvardos, 2016).

Flow deviations by the orography lead to low-level moisture flux convergences over or downstream of mountains (Kottmeier et al., 2008; Kirshbaum et al., 2018). This results in vertical lifting and thus potential convection initiation. Low and high mountain ranges therefore stand out as locally limited hot spots with increased TDs of up to 8. These include the low mountain ranges of Eifel and the Ore mountains in Germany and the Massif Central in France. The hot spot of thunderstorm activity in 300 southwestern Germany is also clearly visible. According to the hypothesis of Kunz and Puskeiler (2010), this hot spot is caused by flow convergence and gravity waves downstream of the Black Forest and Swabian Jura Mountains. Also due to orographic flow modifications, another pronounced maximum in TDs is found along the Spanish-French border linked to the Pyrenees high mountain ranges with up to 8 TDs and flash densities of up to 200.

Hot spots of thunderstorm activity in northern Italy and southern Switzerland at the southern Alps with embedded maxima 305 of more than 12 TDs in Ticino, around Nice, the Po valley and at the Italian-Slovenian border can be seen especially above the southern slopes of the Alps. Reasons for this are predominant southerly air flow from the Mediterranean favoring thunderstorm development in this region (e. g., Manzato et al., 2022). On the Italian peninsula is, additionally to the above mentioned moisture flow, a relation to the mountain ranges is clearly visible: while in the Po Valley in Northern Italy values of around 3 TDs can



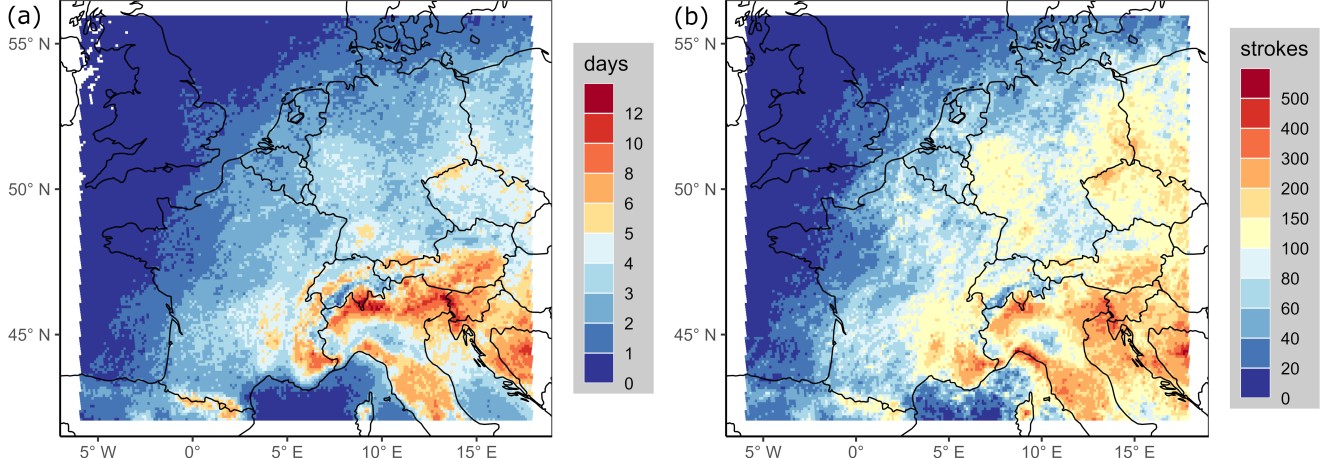

**Figure 6.** Mean annual number of (a) thunderstorm days (TDs) per $10 \times 10\,\mathrm{km}^2$ and (b) lightning density (strokes per $10 \times 10\,\mathrm{km}^2$; EUCLID, 2001–2021, MJJA).

be found, the number increases to more than 8 in the Apennines. In the area around Florence, where the orography is less

pronounced, we observe a local minimum of around 4 TDs. As for the hotspot in the Dinaric Alps, these findings are the result of the combination of warm and moist air masses advected from the Mediterranean Sea and orographic lifting. Kotroni and Lagouvardos (2016) found a positive correlation between the total number of lightning strokes and the SST of the surrounding Mediterranean, which further supports this hypothesis.

Another maximum with a higher number of TDs (at least more than 5) in the Balkans compared to the more northern and

western countries of Europe is located in Bosnia-Herzegovina associated with the higher mountains and ridges of the Dinaric Alps. Another supporting factor for convection are frequently prevailing synoptic conditions with south-westerly flow causing the advection of relatively warm and moist air masses from the Adriatic Sea (Strajnar et al., 2019). The region is also well known for one of the highest convective precipitation totals in Europe (Dietzsch et al., 2017). A less pronounced maximum in the Pozegan Mountains in Croatia with up to 10 mean annual TDs and is also caused by the combination of warm and moist

air masses from the Adriatic Sea and orographic effects.

Minima over the western part of the Mediterranean with less than 1 TD on average are due to the temporal limitation of the lightning data to the months of May to August. The seasonal maximum in this region is towards autumn in September and October (e. g., Kotroni and Lagouvardos, 2016; Piper and Kunz, 2017; Galanaki et al., 2018).

In general, the climatology of lightning density and TDs presented here is largely consistent with other studies (e. g., Wapler,

2013; Anderson and Klugmann, 2014; Poelman et al., 2016; Piper and Kunz, 2017; Rädler et al., 2018; Taszarek et al., 2019; Enno et al., 2020; Manzato et al., 2022). Minor differences are mainly due to different time periods considered, different TD definitions, or different data sources for identifying the thunderstorm distribution.





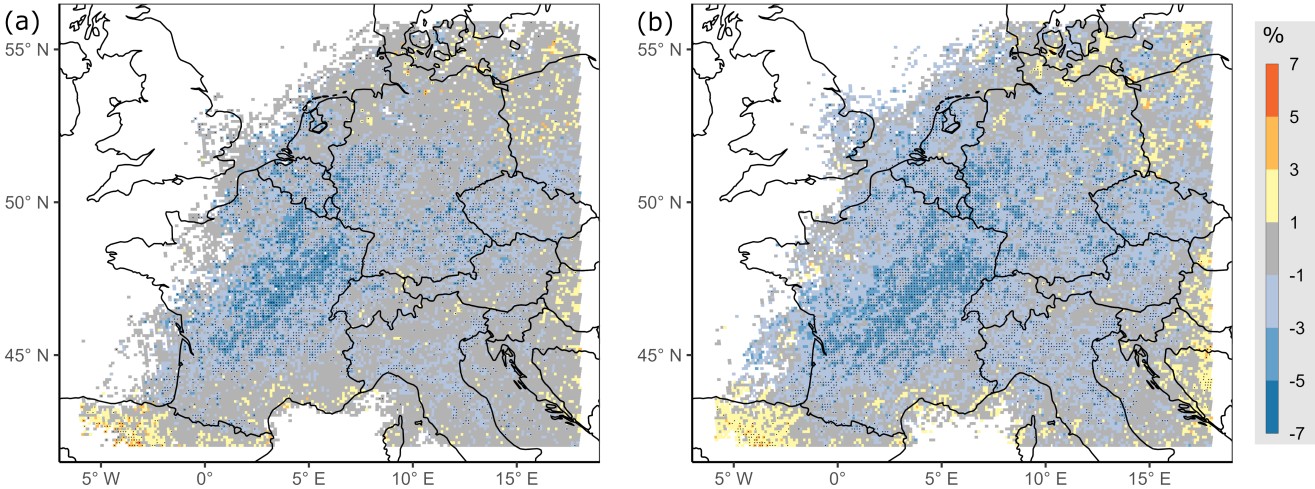

**Figure 7.** Relative trends of (a) TDs and (b) CG strokes per decade. Black dots indicate results of a Mann-Kendall significance test (p = 0.05); grid points with seven or more years without any TDs are excluded (both per $10 \times 10 \, \text{km}^2$; EUCLID, 2001–2021, MJJA).

## 4.2 Trends

A relatively long data time series of 21 years allows us to perform trend analyses based on direct thunderstorm observations
(Fig. 7). As there are too many years without a single TD in some regions, large parts of the North Atlantic, the Mediterranean, and the North Sea as well as Great Britain were excluded from the data and the figure (criterion: at least one TD in 2/3 of the years; see Fig. 6a).

Most noticeable is a large contiguous area with significant negative values down to $-7\%$ for TDs, extending from an area east of the Gironde River in France to the borders of Belgium and up to the Vosges Mountains with further extensions into
Luxembourg and western Germany (Fig. 7a). In contrast, at the southwestern edge of the study area in northern Spain, a smaller contiguous area with positive trends of TDs is found, which is only significant at isolated grid points. Positive trends of TDs are also indicated by a few grid points around the Mediterranean coastal areas, northern Germany, parts of Poland, around the Baltic Sea and in some small areas in the Balkans. However, most of the trends (both TDs and CG stroke density) lack statistical significance.

Trends in CG stroke density show similar spatial structures (Fig. 7b). In the area of France with the strongest negative trends in TDs (see Fig. 7a), the trends for CG stroke density are even more pronounced in terms of strength and statistical significance. This finding suggests a reduction in CG activity per TD as well as in the number of thunderstorms per year.

In total, 36 % of all investigated grid points of TDs show negligible (non-significant) trends of less than $\pm 1\%$ (Fig. 7a, in gray), indicating almost no changes in thunderstorm activity. This finding applies mainly to large parts of Poland and eastern
Germany as well as to the Czech Republic, Austria, Switzerland, Italy, and the Balkan countries contained by the study area.



The finding of a preponderance of negative trends with few positive exceptions is somewhat surprising. Several studies, mainly those based on indirect thunderstorm data, suggested an increase in thunderstorm activity in Europe in recent years. An increase in low-level moisture caused by rising temperatures in response to the Clausius-Clapeyron scaling and, thus, an increase in atmospheric instability has been observed in both radiosoundings (Mohr and Kunz, 2013; Chen and Dai, 2023) and

reanalysis data (e. g. Rädler et al., 2018; Taszarek et al., 2021a). This raises the question of whether the negative trends in the lightning data can be attributed to changes in thunderstorm characteristics or even organizational forms resulting in statistically fewer CG strokes per thunderstorm.

## 5 Analysis of convective clustered events

To investigate possible reasons for the negative trends, especially with respect to thunderstorm characteristics such as organi-

zational forms, approaches other than grid-based methods are needed. Most thunderstorms, especially those with high damage potential, do not only occur as single events, but very often in the form of large convective clusters. By applying ST-DBSCAN (see Sect. 3), we generate a new dataset that is no longer based on artificial metrics such as grid points, but objectively identifies clusters (here CCEs) and thus allows analyses of their spatio-temporal characteristics as well as their annual variability and development.

### 5.1 Spatio-temporal characteristics of convective clustered events

Applying the ST-DBSCAN with the parameters *minPts* = 40, $\varepsilon_{\text{space}}$ = 50 km, and $\varepsilon_{\text{time}}$ = 20 min on the CG strokes data from 2001 to 2021 (MJJA) gives the total number of 46 768 CCEs; i. e., an average of 2 227 CCEs per year (standard deviation ±283). Each CCE is assigned the total number of CG strokes, the area (the area enclosed by the hull polygon; see Sect. 3.2), the duration (defined as the time between the first and the last stroke), and the 3D CG stroke density (defined as strokes per

square kilometer and minute). The correlation between the total number of strokes and the area is 0.87, slightly higher than the correlation between the number of strokes and the duration which is 0.75 (both significant, p = 0.05). In contrast, the correlation between the total number of CG stokes and the CG stroke density is close to zero, showing the large variation of thunderstorm types with different lightning densities within the CCEs themselves.

For both the size and duration of the CCEs the distribution is highly skewed (Fig. 8). As an exemplary illustration, the

characteristics of the CCEs shown in Fig. 5b are as follows: Duration 11.3, 3.4, 6.8, and 1.8 hours and area of 246, 713, 6 620, 35 671, and 4 690 km$^2$ from north to south. The number of CCEs decreases steadily as the size of the CCEs increases. On the other hand for the duration, we see first a sharp increase of the density of less than one hour, with a maximum at about 1.25 hours, and then to the size a fairly similar decrease in the number of CCEs. This striking increase for short-lived CCEs can be attributed to the specification of the density-defining *minPts* and the $\epsilon$-environment in conjunction with the characteristics of

transient, isolated thunderstorm cells, which can have a lifetime of less than one hour. In order to be technically identified as a cluster by ST-DBSCAN, thunderstorm cells (either individually or several thunderstorm cells within the defined $\epsilon$-environment) must have at minimum 40 CG strokes. Therefore, the maximum number of CCEs is slightly shifted towards larger durations



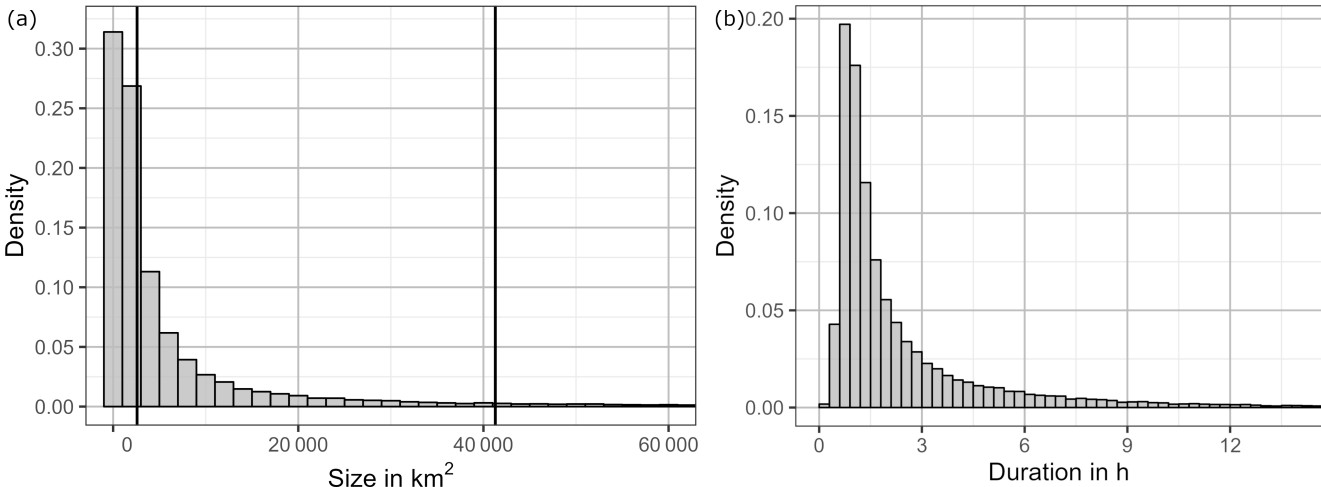

**Figure 8.** Histogram of (a) size and (b) duration of all identified CCEs (EUCLID, 2001–2021, MJJA). For illustration, the vertical black lines in (a) represent the areas of the national territories of Luxembourg (left) and Switzerland (right).

(Fig. 8a). However, this does not contradict the finding of an exponential decrease in the number of CCEs with increasing duration. Only 2.6 % (0.03 %) of the CCEs last longer than 12 (24) hours.

As CCEs are convective active areas considered here as objects, a comparison with the existing literature is difficult as studies with a comparable approach are missing. However, the distribution for the duration of the CCEs is quite similar to the distribution for potential hail tracks derived from radar data in Germany or France (Schmidberger, 2018; Fluck et al., 2021; Mohr et al., 2024). In view of the fact that CCEs may also include several SCSs spatio-temporally close to each other, these results appear to be plausible.

**5.2   Annual variability of CCEs**

The grid-based trend analysis of lightning density and TDs (see Sect. 4.2) revealed predominantly negative trends in west and central Europe. After identifying the CCE event catalog including its main characteristics, the next two sections examine the year-to-year variability of this event catalog in order to identify possible patterns in the temporal distribution and, finally, to find evidence suggesting reasons for the dominant negative trends.

The bagplots (see Sect. 2.5) per year of all CCEs in terms of their size and duration demonstrate a gradual shift towards the occurrence of more smaller CCEs (Fig. 9a) towards the end of the study period, particularly when considering the outer bag. The years at the beginning of the century (blue colors) show more occurrences of larger (and also preferentially longer lasting) clusters than the years of the second decade of the $21^{st}$ century (red colors). For example, the $90^{th}$ quantile of the size of CCEs in 2021 is $22\,900\,\text{km}^2$, which means a reduction of about 31 % compared to $33\,000\,\text{km}^2$ in 2001. In relation to the inner bag of

the bagplots, this can also be seen very clearly. In addition, a year-to-year variability of the distributions in both duration and





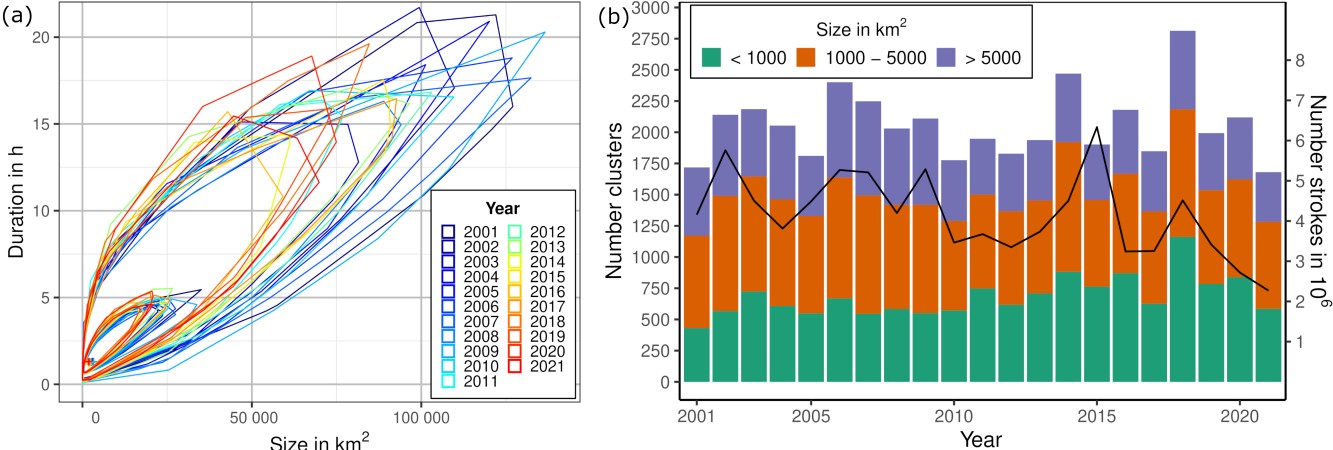

**Figure 9.** (a) Bagplots per year of the CCE catalog with respect to size and duration and (b) annual number of CCEs depending on three size-related categories as stacked colored bars including the total number of CG strokes per year (black line).

size is also visible. This becomes even more evident when looking at the extreme values of the CCE distributions with long duration and large size indicated by the loop.

The total number of CG strokes (Fig. 9b) show a significant (Spearman correlation p = 0.05) negative trend of −0.9 million strokes per decade with the lowest number in 2021 (2.3 million) and the highest number in 2015 (6.4 million). The mean annual number is 4.2 million CG strokes with a standard deviation of about 1.04 million. Dividing the CCE catalog into three size-related categories, it is obvious that there has been an increase in the number of small clusters (< 1000 km, green bars in Fig. 9b), while the number of medium (1000–5000 km², orange bars) and larger clusters (> 5000 km, purple bars) has decreased. Likewise, the ratio of small to large clusters almost doubled over this period (46 % per decade, significant with p = 95 %). However, it should be noted that the number of detected CCEs shows only a weak, non-significant correlation with the number of CG strokes per year ($r = 0.3$, p = 0.065). This is because the CCEs are capable of containing a variety of convective systems, each with a wide range of potential lightning occurrence.

Summarizing these findings under one umbrella: The increase in the number of smaller clusters and the decrease in the number of larger clusters as well as the year-to-year distribution according to the bagplots within the context of the overall CG strokes decrease is an indication for the occurrence of more smaller and spatially separated thunderstorm activity in the second half of the study period. This finding of a change in the mesoscale organization of convective systems could be a reason for the decreasing trends in both TDs and CG stroke density (see Sect. 4.2) as well as the decreasing trend in the total number of CG strokes (Fig. 9b).



## 6 Relationship between thunderstorm activity and NAO

The observed decrease in thunderstorm activity, as evidenced by a reduction of CG strokes and TDs, and the observable change
in thunderstorm organization forms (particularly in terms of the size of the CCEs) motivate an investigation of the potential
influence of large-scale dynamical processes in the atmosphere on these characteristics. Therefore, possible correlations between teleconnection patterns and the TD occurrence as well as their influence on the size and duration of the CCE catalog are
investigated.

The identification of large-scale atmospheric patterns with a significant influence on thunderstorm activity is challenging,
due to the variety of factors that contribute to the formation of thunderstorms. In this study, we present results for the NAO
only. Potential links to other teleconnections such as the East Atlantic or Scandinavian patterns were also investigated, but
they did not demonstrate any notable signal during the study period that could explain the observed changes discussed in the
previous section.

In order to quantify the strength of the relationship between the number of TDs and NAO values, the odds ratio (see Sect. 2.6)
was employed. For the calculation of the OR it is necessary to convert the years into binary values. Therefore, the monthly
mean of the NAO values per year (here MJJA) is determined. Years with a mean value of less than $-0.5$ and a maximum value
of less than 0.75 are classified as $Y_{NAO-}$, those with the opposite sign (mean value of more than 0.5 and minimum value of
more than $-0.75$) as $Y_{NAO+}$. The used mean value of $-0.5$ emerge from the (rounded) $20^{th}$ percentile of the annual mean NAO
values of all the years from the whole time series; the maximum value of less than 0.75 allows the occurrence of months with
weakly positive values. This procedure eliminates the seasonal variability of the thunderstorm activity by comparing annual
values of MJJA periods with each other.

Calculation of the OR of TDs per grid point between $Y_{NAO-}$ and all other years reveals two major pronounced patterns
of contiguous areas with significant OR values above and below one (Fig. 10a). One area with values above one is located
in the north-east of the study area, predominantly over the Baltic Sea and the northeastern parts of Germany and western
Poland, where the probability of TDs occurring during $Y_{NAO-}$ is partially more than twice (OR > 1). The other area with
values below one is located almost exclusively in France, a south-west to north-east broadening band reaching from the north-
western foothills of the Massif Central to the borders of Belgium and Luxembourg and in the east to the outlines of the Vosges
Mountains. In this area the probability of TDs is reduced to almost half (OR < 1).

There are also smaller areas with significant OR values: A smaller band oriented from southwest to northeast along the the
French-Swiss border with values below one, presumably associated with the Alpine foothills. In addition, there is another area
(also with values below one) in the Biscay Bay located directly on the coast of France, just north of the French part of the
Basque country, where the frequency of TDs is almost half as high; and another area with almost doubled frequency of TDs
(positive values) in the border triangle of Austria, Hungary, and Slovakia.

Since the largest contiguous region with relative low OR values, as described above, is found almost exclusively in France,
only CCEs that occur in the majority (i. e., at least 75 % of all flashes of a CCE) in the box shown in Fig. 1 are considered
for the following analysis. During years with $Y_{NAO-}$ values (blue in Fig. 10b), the CCEs occurring in France are smaller in



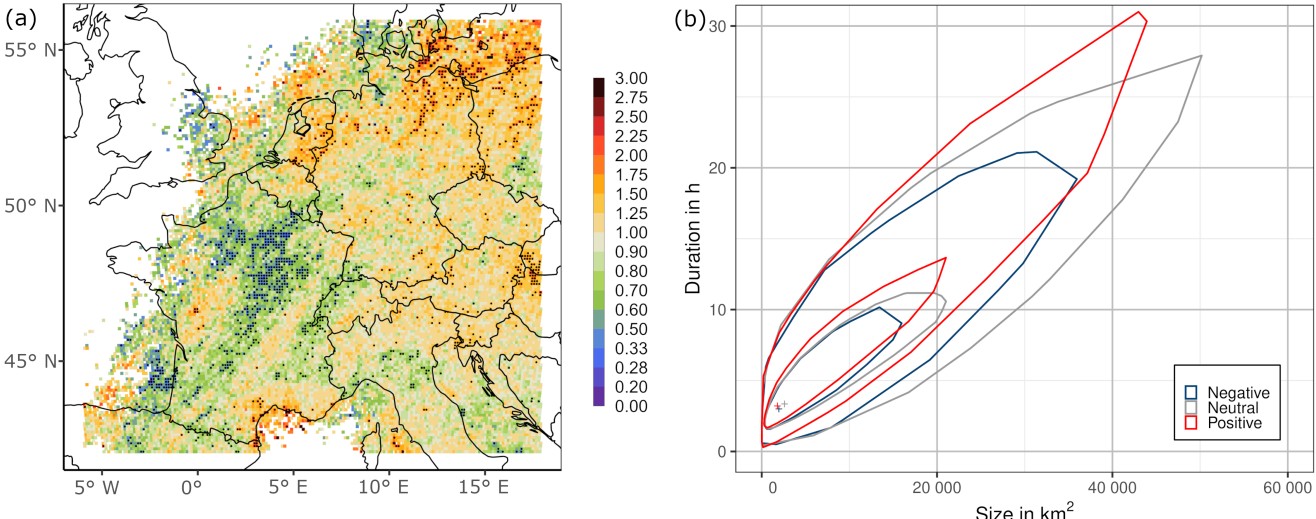

**Figure 10.** (a) Odds ratio between TDs and the negative NAO phase ($Y_{NAO-}$; see definition main text). Areas with more than 7 years without any TDs are excluded (similar to Fig. 7). Black dots indicate the results of a Fisher exact significance test (p = 0.05). (b) Bagplots of the CCEs of differently classified years (based on the NAO values: $Y_{NAO-}$ in blue, $Y_{NAO+}$ in red, and for the other years in gray), which occur mainly in the black box over France (defined in Fig. 1; i. e., at least 75 % of all strokes of an CCE).

size and shorter in duration. Larger and longer lasting CCEs are mainly observed during years with $Y_{NAO+}$ values (in red). In addition, the outer bag of CCEs during $Y_{NAO-}$ years is shifted much further to the left and down compared to the bag of CCEs during $Y_{NAO+}$ or neutral years (in gray). This means that during negative NAO years, CCEs with very large spatial extent and

length occur less frequently than in positive or neutral years.

As described by Folland et al. (2009), the negative phase of the NAO in the summer months July and August goes hand in hand with a mean sea level pressure field where a low pressure system centered over the Shetland Islands causes west-northwest to east-southeast oriented isobars in France. This pressure constellation leads to less advection of convection-favoring warm and humid air masses (cf. Mohr et al., 2019) and therefore to less thunderstorm-favoring environments as well as less

long-lasting thunderstorm systems. The main orientation of radar-based potential hail tracks in France is from southwest to northeast (around 230°; Fluck et al., 2021). As the orientation of the area mentioned above with significant odds ratios indicating a reduction in TDs in France is likewise in this direction, this reduction could be interpreted as the absence of intense thunderstorms. Additionally, the reduced occurrence of typical convection-favoring atmospheric flow patterns with southwesterly flows during negative NAO phases may be the reason for the reduced occurrence of large-sized CCEs during

$Y_{NAO-}$ in France. This hypothesis is also supported by the fact that the area with the most significant negative odds ratio values is located in the lee of the Massif Central (Fig. 10a). The combination of southwesterly flows advecting moist air masses and the orographically favored convection initiation there provides a hot spot for thunderstorms (Fluck et al., 2021). Consequently, a reduction of thunderstorm occurrence therefore should be particularly prominent in the lee of the Massif Central.





The farther east in the study area, the more the typical flow direction in the mid-troposphere is oriented southwesterly during

negative NAO phases (and thus thunderstorm-favoring). This could explain the increase in thunderstorm frequency there during

$Y_{NAO-}$, which is caused by an increase of warm and moist air masses advection to the northeastern part of the study area. As

a result, the values of the odds ratio increase in the northeastern parts of Germany and western Poland and the Baltic Sea.

A detailed analysis of the annual distribution of NAO values (including decomposition into the individual monthly values)

shows that the last decade has an unusual accumulation of phases with negative values. During the period from 2010 to 2019,

seven out of ten years are defined as $Y_{NAO-}$ (Fig. 11). Such a concentration of years with almost exclusively negative values

during MJJA is unique in the entire available NAO data set since the 1950s (not shown). Comparing the area of OR values

below one in France described above (Fig. 10a) with the pronounced area of negative trends in TDs and thunderstorm density

(Fig. 7), there is a large overlap. Considering the accumulation of $Y_{NAO-}$ in the 2010th decade, could be an explanation of the

observed decrease of TDs – caused by an accumulation of years with thunderstorm-reducing flow patterns in conjunction with

negative NAO values.

To summarize all the above findings: the exceptionally often prevailing of negative NAO values during the second decade

of the 21st century likely caused the observed reduction of deep moist convection or lightning activity, respectively (see Fig. 7

and Fig. 10a), and also increased the probability of the occurrence of more smaller and spatio-temporal separated convective

systems (Fig. 10b). This suggests a potential shift in the (mesoscale) organizational forms of the SCSs. It is plausible that these

changes also contribute to the observed reduction in lightning activity.

## 7    Conclusions

This study investigated the spatio-temporal characteristics and changes of thunderstorm activity, as well as the potential causes

related to large-scale teleconnection patterns in western and central Europe. For this purpose, cloud-to-ground (CG) light-

ning strokes from May to August (MJJA) were used from 2001 to 2021 (EUropean Cooperation for LIghtning Detection,

EUCLIDEUCLID). In a first step, grid-based analyses were performed and discussed for CG stroke density and thunder-

storm days (TDs). Linear trend analyses were also conducted for both variables. In a second step, the clustering algorithm

ST-DBSCAN (Spatio-Temporal Density-Based Spatial Clustering of Applications with Noise) from Birant and Kut (2007) was

used to identify temporally and spatially contiguous areas of high convective activity (so-called convective clustered events,

CCEs). This extended the purely grid-based analyses to allow additional examination of the spatio-temporal characteristics of

the CCEs. As part of this process, it was first necessary to develop a statistical method to identify suitable density-defining

parameters for the identification of CCEs. Then, an annual investigation of the spatial and temporal characteristics (in terms

of size and duration) of the CCE catalog was conducted using bagplots (Rousseeuw et al., 1999). Finally, an investigation was

undertaken into the potential influence of large-scale teleconnection patterns such as the North Atlantic Oscillation (NAO) on

the observed trends in CG stroke density, thunderstorm days, and CCEs.

The following important findings can be derived from the results obtained:





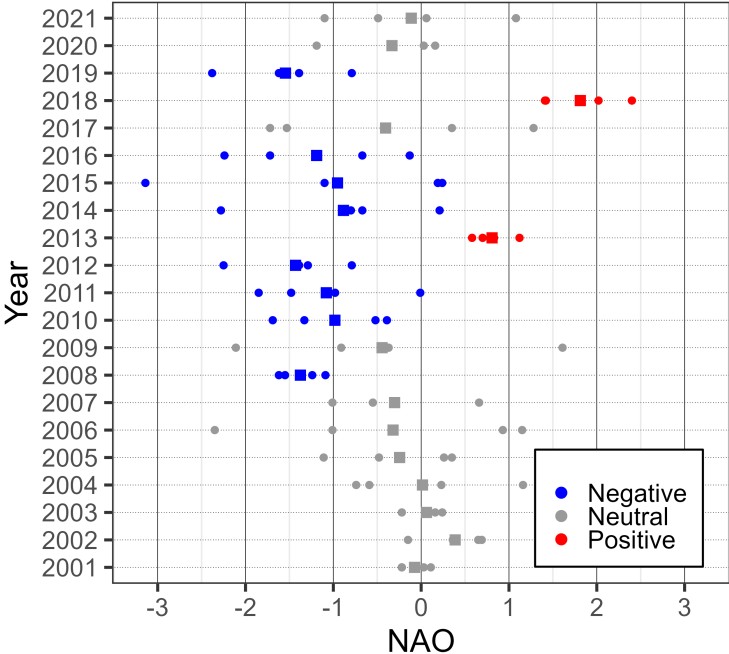

**Figure 11.** Monthly distribution of NAO values per year (2001–2021, MJJA; dots); squares represent the mean value of the respective year. Years with a mean value of less than −0.5 and a maximum value of less than 0.75 are highlighted in blue; in red with reversed sign (mean value of more than 0.5 and minimum value of more than −0.75). Years classified as neither positive nor negative are shown in gray.

1. In contrast to previous studies (e. g., Allen, 2018; Rädler et al., 2019; Taszarek et al., 2021b), most parts of France and Germany show negative trends in both TDs and CG stroke density in this study. Most noticeable is a large contiguous area with significant negative values for TDs down to −7 % in Central France, extending from an area east of the Gironde river in France to the borders of Belgium and up to the Vosges with further extensions into Luxembourg and western Germany. In the area with the greatest decrease of TDs, there was a reduction in CG strokes down to −7 % per decade.

2. Annual investigations of the size and duration of the CCEs found evidence suggesting a trend towards the occurrence of smaller and more spatially separated thunderstorm activity at the expense of larger events, which tend to generate more lightning in Europe.

3. The probability of TDs is nearly doubled in northeastern France in years with an unusual accumulation of months with negative NAO values compared to years without a strong negative accumulation (odds ratio analysis).

4. Combining the first three points leads to the hypothesis that the observed negative trends in thunderstorm days and lightning density in France are plausibly related to an exceptional accumulation of years with negative NAO phases in the last decade, which reduce the probability of thunderstorm days primarily due to a greater proportion of north-westerly



flow components in the above-mentioned region. In addition, these years with negative NAO values predominantly tend
to have smaller and more spatially separated thunderstorm activity.

In conclusion, the findings of our study indicate that the NAO may exert a significant influence on both the TD frequency
and also the mesoscale organization of thunderstorms in western and central Europe. It is hypothesized that a prolonged period
of years with negative NAO values is a major cause for reduced thunderstorm activity, particularly in France.

A significant limitation of the conclusions drawn is the relatively short time frame of 21 years of lightning data. The avail-
ability of longer, homogeneous lightning data (and for the entire European continent) would facilitate the formulation of more
statistically robust statements regarding the relationship between large-scale flow patterns or teleconnections and local-scale
thunderstorm activity. In regions where thunderstorm activity is climatologically rare, monthly totals of TDs (which were the
basic data set for the statistical analyses) are significantly influenced by individual thunderstorms, which may have purely local
causes and are not necessarily related to large-scale flow patterns. Furthermore, the utilization of monthly values for the NAO
represents a relatively coarse description of the large-scale flow pattern. In the future, descriptions of the large-scale atmo-
spheric flow patterns such as weather regimes with a more refined spatio-temporal resolution (Grams et al., 2017) could also
be employed for further investigations.

Since lightning strokes are used in this study as a proxy for deep moist convection, it would also be reasonable to investigate
the effects of microphysical changes on the process of lightning generation. For example, the effects of the investigated in-
crease of the 0°-height (and thus a reduction in the ice-induced charging area)due to rising temperatures or changes in aerosol
concentration could be examined. Additionally, research on the relationship between the thunderstorm type and the lightning
strike intensity, as well as the changes in lightning frequency per individual thunderstorm could prove to be highly insightful.

*Data availability.*   Lightning data from the EUCLID network are not freely available, but in our case were provided by the Blitz-Informationsdienst
from Siemens. Please note: In the meantime, the contact person has changed, so that the data now have to be obtained via the Austrian Light-
ning Detection and Information System (ALDIS). The data of the North Atlantic Oscillation teleconnection index (both monthly and daily)
are freely provided by the US National Oceanic and Atmospheric Administration (https://www.cpc.ncep.noaa.gov/products/precip/CWlink/
pna/nao.shtml, accessed: 13 September 2024).

*Author contributions.*   MA performed all analyses presented in the paper and wrote the initial version. SM and MK wrote the research
proposal and supervised the work. All authors collaborated during the study, discussed the results, and contributed to the review and editing
of the paper.

*Competing interests.*   The authors declare that they have no competing interests.



*Acknowledgements.* This study is the outcome of the subproject 'VarCLuST' of 'ClimXtreme', a project funded by the German Ministry of Education and Research (Bundesministerium für Bildung und Forschung, BMBF) in its strategy 'Research for Sustainability' (FONA) and we sincerely thank the BMBF for funding the two projects (grant number 01LP1901A). We thank the Blitz-Informationsdienst of Siemens (BLIDS, namely Stephan Thern) for providing the lightning data. We also thank Jannick Fischer, Pierre Häsler, Katharina Kuepfer and Mathis Tonn for reading, valuable comments, and suggestions.




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
