# Peer review of "Influence of the North Atlantic Oscillation on annual spatio-temporal lightning clusters in western and central Europe"

_EGUsphere, 2024_

## Referee Comment (RC1)

**Title:** Influence of the North Atlantic Oscillation on annual spatio-temporal lightning clusters in western and central Europe

**Summary:** This article analyzes lightning data collected between 2001 and 2021 in Central Western Europe using a solid statistical method. The analysis was conducted on both gridded data and convective clustered events. The effect of NAO on lightning frequency and convective clustered events was also investigated.

The article is suitable for NHESS. It is well written, and the structure is well crafted. The methodology is robust and clearly explained, with an appropriate level of detail. The figures are informative and comprehensive. The results are noteworthy and original, making the work deserving of publication. However, I have some minor comments that I recommend addressing before publication.

**Minor comments:**

1.  Line 88: suggest using "CCEs" instead of "convective clustered events" for consistency.

2.  Eq. 3: I assume "AR" refers to autoregression. It would be helpful to clarify this in bullet point 3. Additionally, I am not very familiar with the TFPW method—could you expand on how AR(1) is defined and how it was computed in this specific application? Thank you.

3.  Eq. 5: p is not defined, probably is the sample point. Please clarify

4.  Line 166: do border points fail to meet the density criteria because they do not exceed the minPts value? I suggest clarifying this.

5.  Line 245: How did you determine the values [20, 80] and [15, 25]? Earlier, you mentioned 30 km and 20 minutes as initial guesses for parameter selection. Are you selecting parameter ranges at the edges of the strongest gradient region? I assume this is because a strong gradient implies the exclusion of strokes that truly belong to convective clusters. I suggest adding a sentence to further explain the rationale behind choosing these parameter ranges.

6.  Line 264: How is the concavity parameter defined?

7.  Line 270: By "all," do you refer to all clusters detected in the selected example, or in the entire dataset?

8.  Line 272 It appears that the increase in the mean area ratio ends around 2.4. Why do you indicate it ends at 2.2?

9. Line 295: The higher values might also be due to more intense thunderstorms with high electrical activity, though these are rare. Storm initiation in the Adriatic Sea is difficult, but when a storm forms on the surrounding orography and crosses the sea, it often intensifies due to high moisture availability. I have observed this pattern several times in late summer. Stationary storms are not very common over the sea; they typically form near the orography.

10. Line 309: Florence is more inland than the observed local minimum. It looks like is more centered in the "Piana di Pisa", which is the second largest alluvial plain in Italy after the Po Valley.

11. Figure 6: Manzato et al. (2022) found that the maximum lightning density in NE Italy is much higher than in other parts of the Alps. They also used EUCLID data, although for a slightly different period (2005–2019). Your figure shows higher lightning density in Central Italy and Bosnia. Do you have any idea why this discrepancy occurs? Even accounting for the difference in spatial resolution, the variation is difficult to explain. I recommend addressing this point in the manuscript.

12. Line 346-352:  Manzato et al (2025) found a similar results. While in NE Italy instability and moisture is raising according to the radiosoundings data, there is not a significant trend of rainfalls, hailstorms and lightning in the same region. I suggest adding this reference.

    Manzato, A., G. Fasano, A. Cicogna, F. Sioni, and A. Pucillo, 2025: Relationships between Environmental Parameters and Storm Observations in Po Valley: Are They Climate Change Invariant?. *J. Appl. Meteor. Climatol.*, **64**, 267–298, https://doi.org/10.1175/JAMC-D-24-0034.1.

13. Line 401: I suggest avoiding the term "obvious." While a trend exists, it is not particularly strong.

14. Figure 10: It may be helpful to include the black box over France from Fig. 1 in panel (a) of this figure as well.

15. Line 452: I suggest adding a geographic reference for the Shetland Islands. Since they are not shown in Fig. 1a, consider adding "northeast of Scotland."

16. Lines 471-473: computing the correlation between Fig 10a and 7a over France (i.e., between lightning trend and odds ratio) maybe can strengthen your result. I suggest performing this analysis.

17. Line 496: those studies are based on proxies from reanalysis data or soundings. Your study is based on observations of lightnings. I recommend emphasizing this important distinction, which enhances the significance of your findings.

18. Line 525: space missing after the parenthesis

19. Lines 511-514: I recommend explicitly stating that, based on your findings, there appears to be no clear effect of climate change on lightning frequency during the studied period, whereas internal climate variability (specifically NAO fluctuations) has a strong impact. This is an important conclusion. Since the introduction discusses previous studies on thunderstorms and climate change (lines 45–54), a concluding remark on this key result is highly appropriate.

---

## Referee Comment (RC2)

Influence of the North Atlantic Oscillation on annual spatio-temporal lightning clusters in western and central Europe

This manuscript uses lightning observations to identify convective trends in Europe and further investigates the role of the NAO and its anomalous activity on convection. The use of observed lightning data reveals trends that counter proxy-based estimates, raising questions about the compariability of proxies to observations of convection. The article focuses on large-scale outbreaks, that are identified using a spatio-temporal clustering algorithm.

Overall, the work is presented well and cohesively. Some additional discussion of current literature, additional interpretation/reasoning and clarifying aspects are necessary, however.

Below the line-by-line remarks detail the exact locations. More major comments are denoted in bold, where more technical remarks are in regular font.

1. Line 39: Recently, quite a lot of literature on regional climate modeling and convection has emerged, that should be referenced here (Cui et al., 2025, Thurnherr et al., 2025, Brennan et al., 2025, Feldmann et al., 2025, Kahraman et al., 2024)
2. Line 47: Twice "For example" in sentence
3. Line 216: I understand giving the clustering a separate sub-chapter, however, I think it would be beneficial to clarify, that this basically still belongs to the development of methods.
4. Line 305: Nice or Venice? The discussed regions do not match the city mention of Nice…
5. Line 306f: This would benefit from some references / literature support
6. **End of Section 4:** This requires some discussion regarding the length of time periods necessary to robustly identify trends. E.g. Nisi et al., 2018 show no measurable hail trend in Switzerland for 15 years, however, Wilhelm et al., 2024 show a significant positive trend, when evaluating 70 years by exploiting reconstruction techniques. The 20-year period here is still rather short and while significance testing accounts for interannual variability, decadal trends (such as driven by anomalous NAO activity) can still overlay a multi-decadal climate change trend.
7. Line 370: In-text quantifications like this are challenging to visualize as a reader. I would recommend either using a table, if the explicit number should be mentioned, or a visualization with e.g. a bar chart.
8. **Line 371:** The discussion on lifetime deserves a little bit more nuance. While isloated thunderstorms can have a >1h lifecycle, their lightning-active time tends to be shorter than the overall lifecycle. Events with at least 40 CG strikes certainly select towards more intense / larger storms, that tend to have longer lifecycles as well (e.g. Feldmann et al., 2023, Wapler et al., 2017).
9. Line 390: remove more
10. Fig. 9: Please include the information of the displayed percentile either directly in the plot, or in the caption.
11. Line 440: The French-Swiss border region is the Jura, not the Prealps.
12. **Line 455:** This argument on how the negative NAO affects thunderstorm environments should apply to the contradicting proxy studies as well. Some reflection here on where the differences might come from, would be greatly appreciated. Is it differing time periods? Do the proxies not adequately represent convective activity, despite being developed on observational data?
13. Line 478: remove "more" – is it really more storms that are smaller, or is it just a larger fraction of the total storm population, which could still be a smaller absolute number?
14. **End of section 6:** How does lightning activity depend on system size? Does flash rate per km change with system size (see e.g. Feldmann et al., 2023)? Ideally, this aspect would deserve another figure.

How do these results compare to Ghasemifard et al., 2024; where it is concluded that convective trends in Europe are not coupled to synoptic trends?

15. Line 485: double EUCLID
16. **Lines 496ff:** What decades do these studies consider? May this play a role in the comparison?

Recommended literature (partially preprints):

- Cui, R., Thurnherr, I., Velasquez, P., Brennan, K. P., Leclair, M., Mazzoleni, A., T. Schmid, H. Wernli, and C. Schär (2025). A European hail and lightning climatology from an 11-year kilometer-scale regional climate simulation. Journal of Geophysical Research: Atmospheres, 130, e2024JD042828. https://doi.org/10.1029/2024JD042828 (preprint at https://d197for5662m48.cloudfront.net/documents/publicationstatus/231295/preprint_pdf/cb4df177ce3efd8bacbea940c67782f5.pdf)
- Thurnherr et al., 2025: https://d197for5662m48.cloudfront.net/documents/publicationstatus/231295/preprint_pdf/cb4df177ce3efd8bacbea940c67782f5.pdf
- Brennan et al., 2025: https://egusphere.copernicus.org/preprints/2025/egusphere-2025-918/
- Wilhelm et al., 2024: https://nhess.copernicus.org/articles/24/3869/2024/
- Nisi et al., 2018: https://rmets.onlinelibrary.wiley.com/doi/full/10.1002/qj.3286?casa_token=AhPPOv6_4j0AAAAA%3AGEcdWfIIi62f1GmW8-DLUYmrz3nZoNCn2k0NcZgFVrkUisad_y2aKU_DgZBgAYrxU8P1dbxuDOANx4zTrg
- Feldmann et al., 2023: https://www.nature.com/articles/s41612-023-00352-z
- Ghasemifard et al., 2024: https://iopscience.iop.org/article/10.1088/2752-5295/ad22ec/meta
- Kahraman et al., 2024: https://link.springer.com/article/10.1007/s00382-024-07227-w
- Feldmann et al., 2025: https://arxiv.org/abs/2503.07466
- Wapler et al., 2017: https://www.sciencedirect.com/science/article/pii/S0169809516306020?casa_token=vPb7aS5UahwAAAAA:uzj61dUfvGkZR7Nvsr7yAVGFbHaBSPBWjcJu_uJpL5moVuzrMIwDoTDxLu1gj4C0c5dmEqRgRg9L

- Probably too recent, but perhaps still interesting: https://egusphere.copernicus.org/preprints/2025/egusphere-2025-2296/

---

## Author Comment (AC1)

**Reviewer #1**

**General comments**

RC: *This article analyzes lightning data collected between 2001 and 2021 in Central Western Europe using a solid statistical method. The analysis was conducted on both gridded data and convective clustered events. The effect of NAO on lightning frequency and convective clustered events was also investigated. The article is suitable for NHESS. It is well written, and the structure is well crafted. The methodology is robust and clearly explained, with an appropriate level of detail. The figures are informative and comprehensive. The results are noteworthy and original, making the work deserving of publication. However, I have some minor comments that I recommend addressing before publication.*

AR: We would like to thank the anonymous referee for reviewing the manuscript and providing such valuable feedback. We would also like to express our gratitude for the improvements to the manuscript's comprehensibility that this feedback has brought about.

**Specific comments**

RC: *Eq. 3: I assume "AR" refers to autoregression. It would be helpful to clarify this in bullet point 3. Additionally, I am not very familiar with the TFPW method — could you expand on how AR(1) is defined and how it was computed in this specific application? Thank you.*

AR: Thank you for pointing this out. Indeed, "AR" refers here to autoregression. To clarify this, we suggest changing bullet point 3 to:

> Removal of the autoregressive lag-1 process (AR(1), which is the correlation between the time series $y_t$ and the time series $y_{t-1}$) from $y_t$, so called pre-whitening (PW):

RC: *Line 166: do border points fail to meet the density criteria because they do not exceed the minPts value? I suggest clarifying this*

AR: Yes, the border points fail to meet the density criteria. The density criteria expressed in words mean: 'not enough points ($<$ minPts) are within the predefined temporal and spatial neighbourhood.' We would add this explicitly in brackets at the end of the sentence in Line 166:

> $(q_{\text{BorderPts}} < minPts)$

RC: *Line 245: How did you determine the values [20, 80] and [15, 25]? Earlier, you mentioned 30 km and 20 minutes as initial guesses for parameter selection. Are you selecting parameter ranges at the edges of the strongest gradient region? I assume this is because a strong gradient implies the exclusion of strokes that truly belong to convective clusters. I suggest adding a sentence to further explain the rationale behind choosing these parameter ranges.*

AR: Thank you for raising this point. The parameter ranges are indeed determined by the edges of the strongest gradient regions. There is no deterministic algorithm for determining 'optimal' density values. In a one-dimensional case (DBSCAN), therefore, $k$NN-Distance plots are used to determine an optimal threshold. See Fig. 1 for further clarification: on the y-axis, the distance to the fourth-nearest-neigbour (4NN, or more

[Figure]

Figure 1: Illustration of how to determine the 'optimal' density-defining parameters. (a) the distance of each point in b) to the 4th-nearest neigbour in ascending order and the choosen parameter (0.15) to distinguish between cluster and noise. (b) The resulting clusters after applying the DBSCAN-Algorithm with the parameters $minPts$ and $\epsilon_{\mathrm{space}}$.

generally, $k$NN) of all points (x-axis) in ascending order is plotted. The so-called 'knee' of the graph between $y = [0.1,\ 0.2]$ would correspond to the spatio-temporal values $[20, 80]$ and $[15, 25]$ mentioned here. Any value for $\epsilon_{\mathrm{space}} = [0.1, 0.2]$ would generate fairly similar clusters. Lower values (close to 0.1) would transform some border points into noise, and higher values (close to 0.2) would transform some noise into border points. However, this would only result in minor changes to the shape, location, and overall appearance of the resulting clusters.

To summarize this briefly we modified the following sentences from line 244f:

> Using this method, it is now possible to specify a range of values, denoted by $\mathbb{W}$, for the meaningful parameters $\epsilon_{\mathrm{space}}$ and $\epsilon_{\mathrm{time}}$ from these areas with strong gradients: $\mathbb{W}_{\epsilon_{\mathrm{space}}} = [20, 80] \wedge \mathbb{W}_{\epsilon_{\mathrm{time}}} = [15, 25]$. Each combination of parameters obtained from $\mathbb{W}$ then produces clusters with only marginal differences, which should have an insignificant influence on the results presented in later chapters. Additionally, there is only a slight dependence on *minPts*, meaning that $\mathbb{W}$ is relatively independent of the required minimum number of CG strokes.

**RC:** *Line 264: How is the concavity parameter defined?*

AR: The concavity parameter determines whether an 'indentation' into the convex hull around a point cloud is appropriate. It is therefore a relative measure of the concavity of the resulting hull. The value of 1 results in a relatively detailed shape, whereas higher values result in a convex hull (without 'indentations'). This is the basic idea behind the algorithm of Park and Oh (2012) algorithm for determining the depth to which a convex hull should be 'dug' to produce the desired concave hull for a point cloud. The basic concept can be understood by looking at Fig.2: If (length of edge)/(decision distance) > concavity, then a new connection is added to the hull. This process is repeated for all connections at the edge. It continues until none of them satisfy the condition.

[Figure]

Fig. 5. Example of decision distance and digging

Figure 2: The illustration of the process of 'digging' from (Park and Oh, 2012).

We actually think that this description has too much detail to include in the manuscript. Regardless of this, your comment made us realize that we did not apply the initial alpha shapes algorithm by Edelsbrunner et al. (1983), but rather a further development by Park and Oh (2012). Therefore, we will leave the explanation as it is and refer to the correct algorithm in lines 265 and 269 for further details.:

> line 265: An concave hull algorithm Park and Oh (2012), based on the so-called alpha shapes method (Edelsbrunner et al., 1983), line 269: for further details refer to Park and Oh (2012).

**RC:** *Line 272: It appears that the increase in the mean area ratio ends around 2.4. Why do you indicate it ends at 2.2?*

AR: Thank you for pointing this out. Yes, 2.4 is correct here, we changed this accordingly.

**RC:** *Line 295: The higher values might also be due to more intense thunderstorms with high electrical activity, though these are rare. Storm initiation in the Adriatic Sea is difficult, but when a storm forms on the surrounding orography and crosses the sea, it often intensifies due to high moisture availability. I have observed this pattern several times in late summer. Stationary storms are not very common over the sea; they typically form near the orography.*

AR: Thank you for your profound explanation and we would change line 295 to:

> Kotroni and Lagouvardos (2016) argue that this is due to the relatively stationary and isolated thunderstorm systems that tend to occur over the Adriatic Sea. Additionally, thunderstorms that originate due to orography in close proximity could intensify due to the higher moisture content typically found present over the sea. This could result in an increase in intensity and thus also in lightning activity.

**RC:** *Line 309: Florence is more inland than the observed local minimum. It looks like it is more centered in the "Piana di Pisa", which is the second largest alluvial plain in Italy after the Po Valley.*

AR: We thank you for helping with a profound knowledge of the orography in Italy. We will replace the sentence accordingly:

> In the valley of Piana di Pisa (halfway between Florence and the Mediterranean Sea), we observe a local minimum of around 4 TDs.

**RC:** *Figure 6: Manzato et al. (2022) found that the maximum lightning density in NE Italy is much higher than in other parts of the Alps. They also used EUCLID data, although for a slightly different period (2005–2019). Your figure shows higher lightning density in Central Italy and Bosnia. Do you have any idea why this discrepancy occurs? Even accounting for the difference in spatial resolution, the variation is difficult to explain. I recommend addressing this point in the manuscript.*

**AR:** Thank you for pointing out this discrepancy. Having looked into the lightning density per year, we can see that, in 2003, there was a positive anomaly of up to nine thunderstorm days in Central Italy. Additionally, we only used lightning data from May to August and not the whole year as has been used by Manzato et al. (2022). These two factors could explain the discrepancy. Additionally, the non-linear color scale should be considered, as this makes the maximum in NE Italy compared to Manzato et al. (2022) seem visually less pronounced in comparison. We will amend the caption of Figure 6 to explicitly mention the non-linearity of the color scale.

> Note that the color scales are not linear.

**RC:** *Line 346-352: Manzato et al (2025) found a similar results. While in NE Italy instability and moisture is raising according to the radiosoundings data, there is not a significant trend of rainfalls, hailstorms and lightning in the same region. I suggest adding this reference.*

**AR:** Thank you for bringing this work to our attention. It makes an exciting contribution to scientific discourse. Unfortunately, it had not been published by the time this paper was submitted in September 2024. In addition, Reviewer 2 also provided feedback on this point, so we suggest incorporating this paragraph into the text in line 350:

> This underscores the necessity for further discussion: First of all, it is important to note that the lightning data covers only a relatively short period of 20 years. Consequently, the calculated trend may merely reflect decadal variability, which obscures an underlying, more protracted positive trend. Nevertheless, an increase in atmospheric instability has been measurable since the beginning of the 21$^{st}$ century (e. g Battaglioli et al., 2023; Chen and Dai, 2023; Mohr and Kunz, 2013). Therefore, based on the modeling of thunderstorm activity using reanalysis data in the aforementioned studies, an increase should also be reflected in the lightning data. On the other hand, Manzato et al. (2025) showed that there has been no measurable increase in the accompanying phenomena of thunderstorms (hail, heavy rainfall, convective wind gusts) despite a simultaneous increase in atmospheric instability in northern Italy. This raises the question of whether statistical ingredient-based modeling of thunderstorm activity could miss some of the underlying mechanisms of convective storms (Manzato et al., 2025), or whether the negative trends in the lightning data can be attributed to changes in the characteristics or organizational forms of thunderstorms, for example resulting in fewer CGs per thunderstorm.

**RC:** *Lines 471-473: computing the correlation between Fig. 10a and 7a over France (i.e., between lightning trend and odds ratio) maybe can strengthen your result. I suggest performing this analysis.*

**AR:** We thank you for this suggestion. We performed a Spearman correlation test and obtained a value of 0.28 (see Fig. 3 left). This value increases to 0.51, when only significant odds ratio (OR) values are considered (see Fig. 3 right). In both cases, the p-value is below 0.001.

[Figure]

Figure 3: Correlation between OR (all: left; only significant OR: right) and the TD trend. The correlation coefficient equals to 0.28 (left) and 0.51 (right).

However, before concluding that there is only a weak relationship between these variables, it is important to consider the following: This correlation analysis is grid-based, meaning that single, isolated values are given the same importance as grid points embedded in contiguous areas with significant positive or negative OR. For example, Fig. 10a shows that only the region in France stands out as an area with contiguous significant negative values. This indicates that a reduction in thunderstorm activity can be observed over a larger contiguous area of France during NAO-. Additionally, we believe that this analysis is more indicative of a tendency toward a reduction in thunderstorm activity during NAO- in larger regions. Therefore, the correlation of actual values from trends and OR should not be overemphasized.

**RC:** *Line 496: those studies are based on proxies from reanalysis data or soundings. Your study is based on observations of lightnings. I recommend emphasizing this important distinction, which enhances the significance of your findings.*

AR: We thank you for this emphasis and would recommend the following alteration of the sentence in 496:

> In contrast to previous studies based on thunderstorm proxies derived from reanalysis data (e. g. Allen et al. 2018; Raedler et al. 2019; Taszarek et al. 2021), most parts of France and Germany show negative trends in both TDs and CG stroke density (derived from direct lightning measurements) in this study. [...] It should also be kept in mind that the above studies, which are based on reanalysis data, also cover a longer time period.

**RC:** *Lines 511-514: I recommend explicitly stating that, based on your findings, there appears to be no clear effect of climate change on lightning frequency during the studied period, whereas internal climate variability (specifically NAO fluctuations) has a strong impact. This is an important conclusion. Since the introduction discusses previous studies on thunderstorms and climate change (lines 45–54), a concluding remark on this key result is highly appropriate.*

AR: Thank you for mentioning this important issue. We think that additional research is needed to draw such a far-reaching conclusion. It is possible that, in addition to rising temperatures and the associated increase in

instability, climate change also has an impact on the NAO and other factors influencing thunderstorm activity. Nevertheless, we would add the following fifth point in line 511:

> Despite measurable increases in instability caused by global warming, no increase in lightning frequency was observed across large parts of Western and Central Europe during the study period. However, decadal climate variability (here specifically fluctuations in the NAO) seems to have had a measurable impact.

**Technical corrections**

**RC:** *Technical corrections: Line 88: suggest using "CCEs" instead of "convective clustered events" for consistency.; Eq. 5: p is not defined, probably is the sample point. Please clarify; Line 270: By "all," do you refer to all clusters detected in the selected example, or in the entire dataset?; Line 401: I suggest avoiding the term "obvious." While a trend exists, it is not particularly strong.; Figure 10: It may be helpful to include the black box over France from Fig. 1 in panel (a) of this figure as well.; Line 452: I suggest adding a geographic reference for the Shetland Islands. Since they are not shown in Fig. 1a, consider adding "northeast of Scotland.";  Line 525: space missing after the parenthesis*

AR: We thank you for the technical corrections and addressed and incorporated all of them in the text.

**References**

Battaglioli, F., Groenemeijer, P., Púčik, T., Taszarek, M., Ulbrich, U., and Rust, H. (2023). Modeled Multidecadal Trends of Lightning and (Very) Large Hail in Europe and North America (1950–2021). *J. Appl. Meteorol. Climatol.*, *62*(11), 1627–1653. https://doi.org/10.1175/JAMC-D-22-0195.1

Chen, J., and Dai, A. (2023). The atmosphere has become increasingly unstable during 1979–2020 over the northern hemisphere. *Geophys. Res. Lett.*, *50*, e2023GL106125. https://doi.org/10.1029/2023GL106125

Edelsbrunner, H., Kirkpatrick, D., and Seidel, R. (1983). On the Shape of a Set of Points in the Plane. *IEEE Transactions on information theory*, *29*(4), 551–559. https://doi.org/10.1109/TIT.1983.1056714

Kotroni, V., and Lagouvardos, K. (2016). Lightning in the Mediterranean and its relation with sea-surface temperature. *Environ. Res. Lett.*, *11*(3), 034006. https://doi.org/10.1088/1748-9326/11/3/034006

Manzato, A., Fasano, G., Cicogna, A., Sioni, F., and Pucillo, A. (2025). Relationships between Environmental Parameters and Storm Observations in Po Valley: Are They Climate Change Invariant? *J. Appl. Meteorol. Climatol.*, *64*(3), 267–298. https://doi.org/10.1175/JAMC-D-24-0034.1

Manzato, A., Serafin, S., Miglietta, M. M., Kirshbaum, D., and Schulz, W. (2022). A Pan-Alpine Climatology of Lightning and Convective Initiation. *Mon. Weather Rev.*, *150*(9), 2213–2230. https://doi.org/10.1175/MWR-D-21-0149.1

Mohr, S., and Kunz, M. (2013). Recent trends and variabilities of convective parameters relevant for hail events in germany and europe. *Atmos. Res.*, *123*, 211–228. https://doi.org/10.1016/j.atmosres.2012.05.016

Park, J.-S., and Oh, S.-J. (2012). A New Concave Hull Algorithm and Concaveness Measure for n-dimensional Datasets. *Journal of Information Science & Engineering*, *28*(3). https://doi.org/10.6688/JISE.2012.28.3.10

---

## Author Comment (AC2)

**Reviewer #2**

**General comments**

**RC:** *This manuscript uses lightning observations to identify convective trends in Europe and further investigates the role of the NAO and its anomalous activity on convection. The use of observed lightning data reveals trends that counter proxy-based estimates, raising questions about the comparability of proxies to observations of convection. The article focuses on large-scale outbreaks, that are identified using a spatio-temporal clustering algorithm. Overall, the work is presented well and cohesively. Some additional discussion of current literature, additional interpretation/reasoning and clarifying aspects are necessary, however. Below the line-by-line remarks detail the exact locations. More major comments are denoted in bold, where more technical remarks are in regular font.*

**AR:** We would like to express our gratitude to the anonymous referee for reviewing the manuscript and providing valuable comments. Implementing the suggested adjustments has significantly improved this publication.

We would also like to include a slight modification of your statement in our first conclusion point:

> These negative trends, which seem to contradict proxy-based estimates, raise questions about the comparability of proxies with observations of convection.

**Specific comments**

**RC:** *Line 39: Recently, quite a lot of literature on regional climate modeling and convection has emerged, that should be referenced here (Cui et al., 2025, Thurnherr et al., 2025, Brennan et al., 2025, Feldmann et al., 2025, Kahraman et al., 2024)*

**AR:** We thank you for the suggested literature. After looking into it, we agree that it adds additional value and have implemented all of them in the manuscript by adding the following references in line 39:

> ... Cui et al., 2024; Kahraman et al., 2024; Brennan et al., 2025; Feldmann et al., 2025).

Regardless of this addition, we would like to note, that this paper was initially submitted on the 13th of September 2024, before these studies were published.

**RC:** *Line 216: I understand giving the clustering a separate sub-chapter, however, I think it would be beneficial to clarify, that this basically still belongs to the development of methods.*

**AR:** Thank you for mentioning this. As the authors, we also had a lengthy discussion about where to position the clustering chapter. Due to the scope of the topic and the scientific findings that have already emerged from its technical design, we have decided to give clustering its own chapter.

**RC:** *Line 305: Nice or Venice? The discussed regions do not match the city mention of Nice...*

**AR:** Thank you for pointing out this mistake. We have removed 'Nice' from the list and added the following for further clarification:

> ... at the northern slopes of the Po valley ...

**RC:** *Line 306f: This would benefit from some references / literature support*

AR: We thank you for suggesting literature here and included the following references in line 307:

> Nisi et al. 2018; ...

and the following reference in line 310

> (similar to Galanaki et al. 2018)

**RC:** *End of Section 4: This requires some discussion regarding the length of time periods necessary to robustly identify trends. E.g. Nisi et al., 2018 show no measurable hail trend in Switzerland for 15 years, however, Wilhelm et al., 2024 show a significant positive trend, when evaluating 70 years by exploiting reconstruction techniques. The 20-year period here is still rather short and while significance testing accounts for interannual variability, decadal trends (such as driven by anomalous NAO activity) can still overlay a multi-decadal climate change trend.*

AR: That is an important issue and something worth discussing here, thanks for mentioning it. The second reviewer also commented on this, so we suggest adding the following statement in line 350:

> This underscores the necessity for further discussion: First of all, it is important to note that the lightning data covers only a relatively short period of 20 years. Consequently, the calculated trend may merely reflect decadal variability, which obscures an underlying, more protracted positive trend. Nevertheless, an increase in atmospheric instability has been measurable since the beginning of the 21$^{st}$ century (Battaglioli et al., 2023; Chen and Dai, 2023; Mohr and Kunz, 2013). Therefore, based on the modeling of thunderstorm activity using reanalysis data in the aforementioned studies, an increase should also be reflected in the lightning data. On the other hand, Manzato et al. (2025) showed that there has been no measurable increase in the accompanying phenomena of thunderstorms (hail, heavy rainfall, convective wind gusts) despite a simultaneous increase in atmospheric instability in northern Italy. This raises the question of whether the negative trends in the lightning data can be attributed to changes in the characteristics or organizational forms of thunderstorms, for example resulting in fewer CGs per thunderstorm, or whether statistical ingredient-based modeling of thunderstorm activity could miss some of the underlying mechanisms of convective storms (as also discussed by Manzato et al., 2025).

**RC:** *Line 370: In-text quantifications like this are challenging to visualize as a reader. I would recommend either using a table, if the explicit number should be mentioned, or a visualization with e. g. a bar chart.*

AR: We thank you for pointing this out. The exact numbers were added here to give the reader the possibility to visually rank the earlier shown CCEs. But your objection is of course justified. So it might be better to only mention the largest and smallest CCE and their temporal and spatial characteristics here. We therefore changed the sentence in line 370 to:

[Figure]

Figure 1: Illustration of a bagplot and the underlying two dimensional point cloud, from Rousseeuw et al. (1999).

> As an exemplary illustration, the spatial extension of the largest (northernmost, red) and smallest (southernmost, purple) CCE shown in Fig. 5b are 246 713 and 4 690 km$^2$ and with a duration of 11.3 and 1.8 hours respectively.

**RC:** *Line 371: The discussion on lifetime deserves a little bit more nuance. While isloated thunderstorms can have a > 1 h lifecycle, their lightning-active time tends to be shorter than the overall lifecycle. Events with at least 40 CG strikes certainly select towards more intense / larger storms, that tend to have longer lifecycles as well (e.g. Feldmann et al., 2023, Wapler et al., 2017).*

AR: Thank you for raising this point. For our analysis, only the lightning-active time of the overall lifecycle of a thunderstorm is relevant. We would add this explicitly at line 375:

> ..., which can have a lifetime of less than one hour and an even shorter lightning-active period.

**RC:** *Fig. 9: Please include the information of the displayed percentile either directly in the plot, or in the caption.*

AR: We are not entirely sure if it refers to panel (a) on the left, but we assume that this is the case. The so-called 'bags' shown here are not directly percentiles, even though the interpretation is almost identical. But although it is comparable to the box and whiskers of a one-dimensional box plot, they do not directly represent specific percentiles. In fact, these are convex polygons, the outer corner points of which are determined by the underlying points of the point cloud in a more complex way. To get a quick visual impression, refer to Fig. 1 from Rousseeuw et al. (1999): The inner bag contains 50 % of all points, whereas the outer bag is determined by inflating the inner bag by a factor of three (for visibility aspects we use a factor of six). The mathematical precise definition of the so-called bags is more complex, for more details, please refer to Rousseeuw et al. (1999).

**RC:** *Line 455: This argument on how the negative NAO affects thunderstorm environments should apply to the contradicting proxy studies as well. Some reflection here on where the differences might come from, would be*

**greatly appreciated. Is it differing time periods? Do the proxies not adequately represent convective activity, despite being developed on observational data?**

AR: We thank you for pointing this out, and we appreciate getting into this in more detail. We think there are some important points to mention:

Indeed, there is a study considering thunderstorm-relevant parameters and their relationship to the NAO, and they show similar tendencies (speaking of NAO- corresponding with negative anomalies of the equivalent potential temperature $\Theta_e$ over Western Europe, (see. Piper and Kunz, 2017, Fig.13b).

Apart from that, the question of why proxy-based thunderstorm analysis and lightning data analysis seem to differ is an open question and needs to be investigated further. A recently published study investigating the relationship between thunderstorm-related extremes (hail, heavy rainfall, convective wind gusts) and convective parameters also shows differences in their trends (Manzato et al., 2025). One possible explanation would be the one discussed here: Changes in thunderstorm potential (which is captured by reanalysis) do not imply an automatic accompanying increase in the lightning activity, so further differentiation into thunderstorm types (and therefore the lightning intensity per thunderstorm) must be taken into account. This was basically one of the initial ideas to perform a cluster analysis of lightning data to see whether convective active areas (here the CCEs) are showing any trends.

Additionally, it is worth mentioning that proxy studies indicating increasing trends are mostly based on time series that are up to three times longer. Most studies based on reanalysis data do not consider the influence of decadal variability on NAO variability in Europe. This issue has already been addressed at line 350, and we believe that it does not require further discussion at this stage.

RC: *Line 478: remove "more" -– is it really more storms that are smaller, or is it just a larger fraction of the total storm population, which could still be a smaller absolute number?*

AR: Thanks for mentioning this. Yes, it's only a larger fraction of the total storm population. So, 'more' is removed here to be precise.

RC: *Note: Comment 14 has been divided into two separate parts. End of section 6: How does lightning activity depend on system size? Does flash rate per km change with system size (see e.g. Feldmann et al., 2023)? Ideally, this aspect would deserve another figure.*

AR: Thanks for mentioning this issue. During our studies, we also thought about investigating the relationship between flash rate and system size. But there is a major challenge: As can be seen in Fig. 5, for example, the largest CCE (northernmost) contains different thunderstorm systems. Some long-lasting isolated cells (very likely supercells), but also multicellular convection, and also isolated cells with, in comparison to the potential supercells, lower lightning density. Additionally, by construction, larger CCEs tend to encompass larger areas without any lightning. This means a calculated flash rate per area ($\frac{\text{flashes}}{10^3 \; km^2 \; \text{hour}}$) is mainly decreasing with increasing CCE area. But this does not mean that the comprised convective systems themselves have a lower flash rate. In fact, speaking from looking into some individual sample cases, both exist: Large CCEs with an encapsulation without any areas where lightning is absent and have a high flash rate, but in cases of 'unfortunate' spatial locations of the containing convective systems, the flash rate is relatively low. This can also be seen by looking at the distribution of the flash density for different CCE size classes (see Fig. 2). Above a certain size ($> 10\,000 \; km^3$), the distributions of the flash density are almost identical and more or less independent of the size class.

[Figure]

Figure 2: Distribution of the flash density per CCE size classes.

Additionally, there is quite a high correlation between the size and duration of 0.75, but no correlation is present between size and flash density (-0.023). Therefore, it is difficult to draw conclusions about the convective systems (with or without high flash rates) contained within, which is the actual question of interest.

**RC:** *How do these results compare to Ghasemifard et al., 2024; where it is concluded that convective trends in Europe are not coupled to synoptic trends?*

AR: Thank you for highlighting this important issue. This has already been discussed with the authors of the aforementioned study. The conclusion of Ghasemifard et al. (2024) is based on investigations of the relationship between thunderstorm activity and the classification of large-scale flow configurations. In comparison, the NAO is a coarser classification than the ones used by Ghasemifard et al. (2024). Furthermore, changes in the relationship between thunderstorm-related atmospheric parameters (such as those used by Ghasemifard et al. 2024, to model thunderstorm occurrence) and actual lightning occurrence are not considered.

The main conclusion of our paper is rather that an unusual accumulation of NAO- years is a possible explanation for the observed negative trend in Western Europe. Ghasemifard et al. (2024), on the other hand, ask whether positive trends in modeled thunderstorm activity based on reanalysis data can be explained by changes in large-scale synoptic flow configurations. They answer this with 'no'. Therefore, in our opinion, the results presented in our paper should not be viewed as contradictory.

**RC:** *Lines 496ff: What decades do these studies consider? May this play a role in the comparison?*

AR: Thank you for asking this important question. These studies are based on reanalysis data with much longer time series. This is has already been discussed in the additional statement in line 350 so we just refer to this here:

> However, it is important to note that the aforementioned studies, which are based on reanalysis data, cover a much longer time period.

**Technical corrections**

RC: *Line 47: Twice "For example" in sentence; Line 390: remove more; Line 440: The French-Swiss border region is the Jura, not the Prealps.; Line 485: double EUCLID*

AR: We would like to express our gratitude for the resulting improvement in the comprehensibility of the manuscript. We thank you for the technical corrections and addressed and incorporated all of them in the text.

**References**

Battaglioli, F., Groenemeijer, P., Púčik, T., Taszarek, M., Ulbrich, U., and Rust, H. (2023). Modeled Multidecadal Trends of Lightning and (Very) Large Hail in Europe and North America (1950–2021). *J. Appl. Meteorol. Climatol.*, *62*(11), 1627–1653. https://doi.org/10.1175/JAMC-D-22-0195.1

Brennan, K. P., Thurnherr, I., Sprenger, M., and Wernli, H. (2025). Insights from hailstorm track analysis in european climate change simulations. *EGUsphere*, *2025*, 1–29. https://doi.org/10.5194/egusphere-2025-918

Chen, J., and Dai, A. (2023). The atmosphere has become increasingly unstable during 1979–2020 over the northern hemisphere. *Geophys. Res. Lett.*, *50*, e2023GL106125. https://doi.org/10.1029/2023GL106125

Cui, R., Thurnherr, I., Velasquez, P., Brennan, K., Leclair, M., Mazzoleni, A., Schmid, T., Wernli, H., and Schär, C. (2024). A European Hail and Lightning Climatology From an 11-Year Kilometer-Scale Regional Climate Simulation. https://doi.org/10.1029/2024JD042828

Feldmann, M., Blanc, M., Brennan, K. P., Thurnherr, I., Velasquez, P., Martius, O., and Schär, C. (2025). European supercell thunderstorms–an underestimated current threat and an increasing future hazard. *arXiv:2503.07466v1, [physics.ao-ph]*. https://doi.org/10.48550/arXiv.2503.07466

Galanaki, E., Lagouvardos, K., Kotroni, V., Flaounas, E., and Argiriou, A. (2018). Thunderstorm climatology in the mediterranean using cloud-to-ground lightning observations. *Atmos. Res.*, *207*, 136–144. https://doi.org/10.1016/j.atmosres.2018.03.004

Ghasemifard, H., Groenemeijer, P., Battaglioli, F., and Púčik, T. (2024). Do changing circulation types raise the frequency of summertime thunderstorms and large hail in europe? *Environ. Res.: Climate*, *3*(1), 015008. https://doi.org/10.1088/2752-5295/ad22ec

Kahraman, A., Kendon, E. J., and Fowler, H. J. (2024). Climatology of severe hail potential in europe based on a convection-permitting simulation. *Clim. Dynam.*, *62*(7), 6625–6642. https://doi.org/10.1007/s00382-024-07227-w

Manzato, A., Fasano, G., Cicogna, A., Sioni, F., and Pucillo, A. (2025). Relationships between Environmental Parameters and Storm Observations in Po Valley: Are They Climate Change Invariant? *J. Appl. Meteorol. Climatol.*, *64*(3), 267–298. https://doi.org/10.1175/JAMC-D-24-0034.1

Mohr, S., and Kunz, M. (2013). Recent trends and variabilities of convective parameters relevant for hail events in germany and europe. *Atmos. Res.*, *123*, 211–228. https://doi.org/10.1016/j.atmosres.2012.05.016

Nisi, L., Hering, A., Germann, U., and Martius, O. (2018). A 15-year hail streak climatology for the alpine region. *Q. J. R. Meteorol. Soc.*, *144*(714), 1429–1449. https://doi.org/10.1002/qj.3286

Piper, D., and Kunz, M. (2017). Spatiotemporal variability of lightning activity in europe and the relation to the north atlantic oscillation teleconnection pattern. *Nat. Hazards Earth Syst. Sci.*, *17*(8), 1319–1336. https://doi.org/10.5194/nhess-17-1319-2017

Rousseeuw, P. J., Ruts, I., and Tukey, J. W. (1999). The bagplot: A Bivariate Boxplot. *The American Statistician*, *53*(4), 382–387. https://doi.org/10.1080/00031305.1999.10474494